# Tailoring the Barrier Properties of PLA: A State-of-the-Art Review for Food Packaging Applications

**DOI:** 10.3390/polym14081626

**Published:** 2022-04-18

**Authors:** Stefania Marano, Emiliano Laudadio, Cristina Minnelli, Pierluigi Stipa

**Affiliations:** 1Department of Science and Engineering of Matter, Environment and Urban Planning, Marche Polytechnic University, 60131 Ancona, Italy; e.laudadio@staff.univpm.it (E.L.); p.stipa@staff.univpm.it (P.S.); 2Department of Life and Environmental Sciences, Marche Polytechnic University, 60131 Ancona, Italy; c.minnelli@staff.univpm.it

**Keywords:** PLA, barrier properties, food packaging, nanoconfinement, biocomposites, clay nanoparticles, copolymers, molecular dynamics

## Abstract

It is now well recognized that the production of petroleum-based packaging materials has created serious ecological problems for the environment due to their resistance to biodegradation. In this context, substantial research efforts have been made to promote the use of biodegradable films as sustainable alternatives to conventionally used packaging materials. Among several biopolymers, poly(lactide) (PLA) has found early application in the food industry thanks to its promising properties and is currently one of the most industrially produced bioplastics. However, more efforts are needed to enhance its performance and expand its applicability in this field, as packaging materials need to meet precise functional requirements such as suitable thermal, mechanical, and gas barrier properties. In particular, improving the mass transfer properties of materials to water vapor, oxygen, and/or carbon dioxide plays a very important role in maintaining food quality and safety, as the rate of typical food degradation reactions (i.e., oxidation, microbial development, and physical reactions) can be greatly reduced. Since most reviews dealing with the properties of PLA have mainly focused on strategies to improve its thermal and mechanical properties, this work aims to review relevant strategies to tailor the barrier properties of PLA-based materials, with the ultimate goal of providing a general guide for the design of PLA-based packaging materials with the desired mass transfer properties.

## 1. Introduction

The increasing use of petroleum-based plastics has raised serious environmental issues closely related to their resistance to biodegradation. In the specific case of materials used in packaging, food-contaminated plastics cannot be recycled, so significant amounts of non-degradable material are constantly accumulating in the natural environment and landfills [1]. In this context, growing research interest is directed toward a greater use of renewable resources for the production of bio-based polymers with certain desired functionalities that are, at the same time, fully biodegradable and recyclable [2]. Among a large number of biopolymer candidates, environmentally friendly thermoplastic polylactide (PLA), derived from renewable agro-resources, has certainly attracted much attention due to its promising attributes for packaging applications. These include biotic and non-biotic degradability, clarity, stiffness, low-temperature heat sealability, GRAS (Generally Recognized as Safe) status as well as suitable barrier properties to flavors and aromas, which have made PLA the most widely produced bioplastic at present for certain biological food contact applications [3]. For instance, Biota^TM^ PLA-bottled water, Noble^TM^ PLA-bottled juices, and Dannon^TM^ yogurts are the main PLA-based packaging examples available on the market.

In this context, however, it should be noted that not all PLA grades are suitable for packaging applications, as the properties of PLA strongly depend on the physical state of the polymer, which in turn is primarily influenced by the stereochemistry, composition, and molecular weight of PLA. In fact, lactide (the cyclic dimer of the chiral lactic acid moiety) exists in three diastereoisomeric forms such as l-lactide, d-lactide, and meso-lactide, whose polymerization via a ring-opening reaction (ROP) affords several different PLA grades [4]. Optically pure PLA such as isotactic poly(l-lactide) (PLLA) and poly(d-lactide) (PDLA) are crystalline in nature with a melting point around 180 °C, while atactic poly-(meso-lactide) (PDLLA) is a fully amorphous material with a glass transition temperature of 50–57 °C. Depending on the ratio of optically active l- and d,l-monomers, it is possible to tune the amorphous and crystalline content of PLA and thus its properties. Detailed information on this subject can be found by interested readers in [5,6]. In addition, it has been widely reported that other factors such as the chain orientation and crystal packing of PLA also affect the crystallinity content, crystal thickness, spherulite size, and morphology, which in turn can strongly influence the final polymer properties [7]. 

PLA samples with the highest content of amorphous regions and low molecular weight (MW) are not used in packaging as they are less thermally stable and degrade much faster than their enantiomerically pure counterparts with high MW PLA (PLDA or PLLA) [8]. However, as shown in Table 1, even the highest performing PLA grades with the highest degree of crystallinity (typically PLA with 96–99% l-lactide) are not good enough to extend the applicability of PLA in packaging. They show relatively poor barrier performance as well as poor heat resistance and brittle fracture behavior compared to conventional petroleum-based plastics [9,10,11]. In terms of mass transfer properties, PLA exhibits moderate permeability (P) values for oxygen (O_2_) and carbon dioxide (CO_2_), which are higher than those of polyhydroxy butyrate-co-valerate (PHBV), polyvinylidene chloride (PVDC), ethylene vinyl alcohol (EVOH), polyethylene terephthalate (PET), and polyvinyl alcohol (PVOH), but mostly lower than those of the other polymers. However, with the exception of PVOH and EVOH, PLA has a very high P_Water_ compared to the other polymers, which also limits applications that require a high barrier to water vapor.

While most recent review articles have focused on the functional properties of PLA materials (i.e., mechanical and thermal properties) [12,13,14,15,16], there are no studies that have provided a comprehensive understanding of PLA barrier performance and reported the available methods to improve it. Therefore, this review provides a critical overview of recent strategies to improve the barrier properties of PLA-based plastics, focusing on the approaches that have the least environmental impact. These include crystallization and orientation, crystal modifications, blending with other impermeable biopolymers and/or nanofillers, and co-polymerization. In addition, this review presents several models for predicting the permeability of PLA-based plastic films through a molecular dynamics simulation approach (MD). To provide an overview of the strategies used by researchers, this review begins with a brief introduction to the key parameters used to characterize gas transport properties in polymer films and ends with a summary and outlook on the design of PLA-based packaging materials with the desired mass transfer properties.

**Table 1 polymers-14-01626-t001:** Comparison of the main barrier (permeability coefficient for O_2_, water, and CO_2_ in $\mathrm{Kg}\cdot m\cdot m^{-2}\cdot s^{-1}\cdot {\mathrm{Pa}}^{-1}$) and the thermal and mechanical properties of polymers used in food packaging.

Polymer *	Crystallinity Content (%)	P_O2_ ^a^	P_Water_ ^b^	P_CO_2__ ^c^	T_deg_ ^d^ Onset (°C)	Tensile Strength (MPa)	Young Modulus (GPa)	Elongation at Break (%)	Ref.
PLA	0–40	310 ± 150	161 ± 41	2811 ± 842	270	20–70	3.1–4.8	3.6–8.8	[9,12,13,17,18,19,20]
PHBV	40–60	1.1–3.2	15–24	14–40	230	35–40	3.6–5.2	4–970	[21,22,23]
PP	30–60	1790	312	10,500	335–450	31–48	0.2–1.4	550–1000	[20,24,25]
PET	17–40	35.9	7.8	35.9	406	45	2.7–4.1	335	[20,24,25,26,27,28,29]
PVC	10	449	16.5	247	250	4–23	2.7–3.0	200–240	[20,30,31,32,33]
PVDC	40–50	0.1–0.3	1.2–7.3	2.3–12	130	25–110	1.2–1.8	30–80	[34,35,36]
PVOH	15	0.7–9.5	430–840	18.2	200	31	0.08–0.7	57–122	[20,37,38,39,40,41,42]
EVOH	58–70	0.5–7.1	320–560	5.1–14.3	397	55–65	0.4–1.2	100–225	[38,43,44]
LDPE	47	3100	5.5	18,600	395	33	0.3–0.6	1075	[20,45,46,47,48,49]
HDPE	74	424	2.1	538	389	16–21	0.5–1.2	10.7–13.7	[20,48,50,51]

* PHBV, polyhydroxy butyrate-co-valerate; PP, polypropylene; PET, polyethylene terephthalate; PVC, polyvinyl chloride; PVDC, polyvinylidene chloride; PVOH, polyvinyl alcohol; EVOH, ethylene vinyl alcohol; LDPE, low-density polyethylene; HDPE, high-density polyethylene. ^a^ Oxygen permeation coefficient (P): $P\times 10^{-20}\frac{\mathrm{Kg}\;m}{m^{2}s\;\mathrm{Pa}}$; ^b^ Water vapor permeation coefficient (P): $P\times 10^{-16}\frac{\mathrm{Kg}\;m}{m^{2}s\;\mathrm{Pa}}$; ^c^ Carbon dioxide permeation coefficient (P): $P\times 10^{-20}\frac{\mathrm{Kg}\;m}{m^{2}s\;\mathrm{Pa}}$; ^d^ Onset of degradation temperature measure by thermogravimetric analysis.

## 2. Concept of High Gas Barrier Material: Fundamentals of Permeation and Diffusion

The term ‘barrier’ refers to the inherent ability of a material to allow the exchange or permeation of low molecular weight chemical species such as gases, water vapor, and certain organic compounds (aroma molecules) [52]. This capability is extremely important in the food packaging industry, as the most important function of any packaging system is to maintain the quality and safety of the contents. In fact, foods are chemically unstable by nature and therefore need to be protected from various spoilage possibilities, lipid oxidation, and microbial contamination being the main causes of their deterioration [53]. Therefore, polymeric materials intended for use in many packaging applications must form a “high barrier” (i.e., they must prevent the penetration of substances from the packaging environment into the food and vice versa) as much as possible. The gases typically involved in food packaging are oxygen, water vapor, and carbon dioxide, and the corresponding permeability rates are known as O_2_TR, WVTR, and CO_2_TR, respectively. 

It is well-known that permeation of low molecular weight chemicals through a nonporous polymer matrix occurs via a combination of two processes (e.g., solution and diffusion). Figure 1 shows that gas molecules are first dissolved on one side of the polymer film, followed by molecular diffusion to the other side (postulated by Thomas Graham in 1866) [54]. These processes, can therefore be described by a simple solution–diffusion mechanism using Henry and Fick’s laws, which can be formally expressed in terms of permeability P, solubility S, and diffusion D, according to Equation (1):(1)$$P=D\cdot S=\frac{J\cdot d}{\Delta_{p}}=\frac{amount\;\cdot \;material\;thikness}{surface\;area\;\cdot \;time\;\cdot \;pressure\;difference}=\frac{[{\mathrm{cm}}^{3}]\;\left(SATP\right)\;\left[\mathrm{cm}\right]}{[{\mathrm{cm}}^{2}]\;\left[s\right]\;\left[\mathrm{Pa}\right]}\;$$
where J is the amount of material transported per unit time through a unit area with thickness d at standard ambient temperature and pressure (SATP; 298.15 K and 10^5^ Pa) and $\Delta_{p}$ is the constant partial pressure difference between both sides of the polymer matrix [55]. Therefore, the magnitude of permeability is determined by the diffusion rate (D), which is a kinetic parameter, and the solubility (S), a thermodynamic parameter related to the amount of permeate sorbed by the polymer membrane. In the specific field of packaging, it is worth noting that the permeability *p* values at different locations in a package can vary greatly and are only an approximate estimate of the actual overall permeability. This could be attributed to different material thicknesses of the walls and seals, multilayer compositions, and/or the presence of defects (i.e., pores or leaks). Therefore, it is critical to more accurately calculate the overall permeability Q of the package using the flux J, according to Equation (2) [36]: (2)$$Q=\sum_{i}^{n}P_{i}d_{i}=\sum_{i}^{n}A_{i}\;J_{i}$$

Q is thus the sum of the permeability values $P_{i}$ for each individual packaging component i relative to its wall thickness $d_{i}$. Based on Equation (1), this is equal to the sum of the corresponding flux $J_{i}$ multiplied by the surface area $A_{i}$ of the component. In practice, there are several methods for measuring the permeability of plastics in the form of films, sheets, laminates, co-extrusions, or plastic-coated materials, all of which are published by standards organizations such as the American Society for Testing Materials (ASTM International) and the International Organization for Standardization (ISO) [56]. These include the isostatic method [20] (also known as the continuous-flow method) and the quasi-isostatic method [57] (also known as the lag-time or constant-volume/variable-pressure method). In the latter method, a polymer film is exposed to the permeant on one side and the concentration is accumulated to values below 5 wt% on the other side. The samples are quantified at specific time intervals to produce a graph showing the amount of permeant versus time. The intercept of the x-axis is taken from the steady-state portion of the graph. This is the lag time ($t_{\theta }$) used to estimate D as follows:(3)$$\;t_{\theta }=\frac{L^{2}}{6D}$$
when the permeation is in a steady-state, P can be estimated from the slope of the linear part of the permeation plot. 

For semicrystalline polymers (e.g., PLA), transport properties are generally evaluated using a two-phase model that identifies an impermeable crystalline phase and a permeable amorphous matrix [58]. According to this model, sorption can only occur in the amorphous regions as the denser crystalline organization makes it difficult for the permeant molecules to reach the sorption sites due to limited mobility. However, deviations from this simplified model have been reported [59,60,61,62,63] because, in addition to crystallinity/amorphous fraction, permeation can be influenced by other intrinsic and/or extrinsic factors such as crystal architecture, polarity, polymer microstructure, chain packing, amorphous phase morphology, presence of additives, and environmental conditions (i.e., temperature and relative humidity) [64,65].

In light of developing highly efficient and economically viable PLA-based packaging materials, the main approaches to optimize the transport properties of PLA including molecular dynamics simulations are presented below to identify the main factors that determine the permeability performance of the material.

## 3. PLA Morphology Modifications

### 3.1. Degree of Crystallinity and Crystal Polymorphism

It is well-established that the degree of crystallinity affects a variety of polymer properties including gas and/or water vapor permeability behavior. In fact, the crystalline phase is highly ordered, aligned, and denser than the amorphously oriented phase, which does not exhibit repeating patterns in the solid state. Common film-forming polymers used in packaging are semi-crystalline polymers (e.g., PET, PP, polyethylene (PE), PVDC, polyamide (PA), and EVOH), in which both the amorphous and crystalline regions coexist within the polymer matrix. Since gas diffusion through polymers is primarily controlled by the packing mode of the molecular chain segments, it is generally assumed that ordered crystalline domains should act as an effective barrier to the diffusion of gases and small molecules, making the amorphous phase the only pathway available for permeation [66]. Moreover, penetrants cannot sorb in crystalline structures because their solubility coefficients are lower compared to those of their amorphous counterparts [67,68]. For this reason, a high degree of crystallinity is particularly desirable for polymers intended for the food packaging industry [69,70]. Therefore, in a similar way to many other semicrystalline polymers, an increase in the degree of crystallinity in PLA should result in a decrease in the permeability of most low molecular weight compounds. As previously mentioned, the degree of crystallinity of PLA can be easily tuned by polymerizing a controlled mixture of the l-, d-, and meso-lactide. Depending on the isomer ratio, PLA can be fully amorphous or semicrystalline and the more optically pure polymers display higher crystallinity fractions because of higher chain symmetry. In particular, the degree in crystallinity ($X_{c}$) increases as l-lactide increases, except for PLA 50% l-lactide and 80% l-lactide, which both present 0% of $X_{c}$ [9]. Alternatively, a higher degree of crystallinity can also be obtained by post-processing corona treatment and drawing (i.e., uniaxial or biaxial orientation of amorphous PLA samples) [71,72,73]. $X_{c}$ in polymers is commonly measured using the differential scanning calorimetry (DSC) technique by dividing the enthalpy of fusion of the studied samples with the reference enthalpy value for 100% crystalline PLA (93 J/g) [12]. The gas permeability performance of several PLA grades as a function of crystallinity content has been widely investigated over the last couple of decades, and a large body of data can be found in the literature in this regard [73,74,75,76,77,78,79]. However, contrary to expectations, there did not appear to be a clear relationship between PLA barrier performance and its degree of crystallinity. In particular, the decrease in the gas permeability and water sorption did not occur linearly or specifically, not to the expected extent, with increasing crystallinity of PLA. For example, Tsuji and coworkers [78] conducted an in-depth investigation of the effect of the $X_{c}$ of various PLA films on their water vapor permeation coefficient (P_water_). PLA raw polymers were supplied or synthesized by ROP and the resulting samples were solution-cast to form thin films (thickness of ~50 µm). Films were subsequently made amorphous by melt quenching and recrystallized at different time and temperature to obtain samples with $X_{c}$ ranging from 0 to 35%. Results showed (summarized in Table 2) that the P_water_ of PLLA films decreased monotonically from 2.18 to 1.14×10^14^ Kg/m/m^2^/s/Pa as a function of increasing $X_{c}$ from 0 to 20%. However, at higher $X_{c}$ a plateau in the P_water_ values was reached, reporting no further reductions.

Similarly, Drieskens et al. [74] subjected compression-molded (amorphous) PLA samples to cold crystallization at different temperatures and times to obtain samples with various $X_{c}$ and morphologies. They found that at low level of crystallinity (~30–35%), the oxygen permeability coefficients decreased almost linearly with increasing crystallinity, while at higher $X_{c}$ (>40%), the opposite trend was observed. An analogous observation was reported in the work conducted by Guinault et al. [75], whereby the analysis of measured oxygen and helium permeability values showed that for crystallinity degrees higher than around 35%, the diffusion coefficient increased with increasing $X_{c}$, confirming the poor relationship between crystallinity and barrier properties for PLA. As a last example, Colomines et al. [79] obtained, from oxygen and helium permeation measurements, comparable permeability coefficients between amorphous PLA and semicrystalline PLA film samples prepared by compression molding, showing that the degree of crystallinity does not appear to have any effects on PLA permeability behavior. 

These studies and other similar ones suggest that the barrier properties of PLA might be less dependent on the crystalline content than expected and the effect of the resulting post-processing PLA microstructures should be further explored in relation to their different permeability behaviors. In this context, some investigations have attempted to find a correlation between the barrier properties and potential PLA crystal structure modifications. In fact, depending on processing conditions, PLA can crystallize in up to four polymorphs: α, β, γ, and the more recently reported α′ (or otherwise known as δ) form [80]. Among all, the α form, obtained by simple crystallization from the melt at high temperature, is the most stable polymorph. β and γ forms are obtained in more extreme or special conditions such as employing high energy stretching of the α form at high temperature (β form) or conducting crystallization on the hexamethylbenzene substrate (γ form) [81]. However, the α′ form reported by Zhang et al. [82] was obtained at low crystallization temperature and the resulting crystal structure was found to differ only slightly from the α structure. By simply changing the crystallization temperature from low to high values, it is therefore possible to obtain samples containing PLA in either the pure α form or α′ form, or a mixture of the two polymorphs in the same system. Differences between the α and α′ forms lie only in their chain packing mode, whereby the larger lattice dimension and the weaker interchain interactions make the α′ form somewhat more disordered compared to the α form. This peculiar difference in the packing conformation between the two crystal microstructures may affect the barrier properties of processed PLA samples. Cocca et al. [83] undertook this investigation by subjecting a series of compression-molded PLA samples to different crystallization temperatures ranging from 85 up to 165 °C. Based on the wide angle X-ray (WAXD) results, the heating treatment afforded PLA samples with different α/α′ ratios: samples in the pure α form (T_c_ > 145 °C), samples in the pure α′ (T_c_ < 95 °C) form, and samples containing a mixture of both crystal structures (105 °C < T_c_ < 125 °C). Figure 2A–C shows the estimated fractions of the α form in PLA compression-molded films and the related optical micrographs after cold crystallization as well as the corresponding water vapor permeability of crystallized films as a function of $X_{c}$. The water vapor permeability behavior of PLLA films showed a clear dependence on the crystal conformation. In particular, samples uniquely containing the α′ form showed the maximum permeability value, which started to progressively decrease with larger ratios of α to α′ content until the minimum value was reached for samples containing only the α form. Similar results were also observed in another following study [75]. This clearly indicates that the molecular packing mode strongly influences the permeability behavior of PLA and caution should be paid in choosing appropriate crystallization conditions to favor the formation of the more impermeable α crystal structure.

### 3.2. Amorphous Phase Conformation

Besides the crystallinity content and crystal conformations, the amorphous phase dynamics and particularly its degree of coupling with the crystalline phase have been reported to play an important role in the barrier properties of many polymeric materials [68,84,85,86,87]. In fact, as displayed in Figure 3, the amorphous regions in semicrystalline polymers (i.e., PLA) are typically affected by constraints imposed by the crystalline regions, which identify two different amorphous phases: the “free” mobile amorphous fraction (MAF) and the more rigid amorphous fraction (RAF) [88,89,90].

The latter does not relax as it is confined by crystalline lamellas in such a way that the amorphous phase chain mobility is greatly reduced. This creates an excess of free volume attributed to a de-densification effect [91]. For this reason, this particular constrained amorphous region appeared to have a more important role on PLA permeability behavior compared to the crystalline fraction content [61,74,77,87]. For example, Drieskens et al. [74] investigated the potential correlation between the morphology of isotropic PLA (modified by cold crystallization at different temperatures and times) and changes in PLA oxygen transport characteristics (permeability, diffusion and solubility). Amorphous PLA samples were obtained by direct quenching from the melt and subsequently annealed at the corresponding crystallization temperature (T_c_) for different times (from 10 min up to 24 h). Following isothermal crystallization, different PLA crystal morphologies were reported, and the resulting microstructures characterized using DSC, electronic, and optical microscopy. In accordance with previous results, at low crystallinity increment, gas permeability was reduced due to an increase in the number and size of crystals, which in turn, increased the tortuosity of the transport path. However, at higher crystallinity contents, when the polymer matrix was completely filled with crystals, the glass transition fully shifted to higher temperatures, indicating that a constrained amorphous phase (RAF) was formed during PLA crystallization. By measuring the thermodynamic gas solubility component, it was observed that samples containing increasing amounts of the RAF fraction showed much higher gas solubility values. This was ascribed to be the major cause of plateauing in the permeability behavior at higher crystallinity levels. In line with these results, Guinault and coworkers [77] undertook an in-depth study to further elucidate the relationship between gas properties, crystallinity (ranging from ~2 to 61%), and RAF content (ranging from ~0 to 20%) in twenty PLA film samples. The permeability performance was analyzed using two model molecular probes: oxygen (‘red sea mechanism’ transport) and helium (‘fluid like mechanism’ transport). In the first mechanism, the gas transport is dependent on the volume accessible based on the microstructure of the polymer matrix, while the second one, having a much smaller size, behaves more like a “liquid” through the polymer microvoids. In both the PLLA and PDLA samples, a significant drop in helium permeability was observed upon crystallization of the fully amorphous extruded samples. However, when oxygen transport was analyzed, which is more sensitive to the polymer microstructure, there was only a small reduction in the permeability with increasing crystallinity up to approximately 35% for both sample groups. At higher crystallinity levels, the permeability to oxygen was observed to increase as a function of increasing RAF fraction. This anomalous behavior was ascribed to an increase in the overall free volume due to poor coupling between RAF and the crystalline fraction. This observation was also found in accordance with a previous study conducted by Del Rio et al. [92], whereby the evolution of the free volume was measured upon annealing of an amorphous PLLA sample using positron annihilation lifetime spectroscopy (PALS). This technique allowed for the identification of an increase in the free volume within the annealed PLLA matrix due a change in conformation of the amorphous regions occurring during crystallization: from a folded or coil conformation in the quenched samples containing exclusively MAF to a more open conformation in samples containing greater content of RAF and crystalline fraction. While this change in conformation caused a decrease in the average hole size from 95.7 to 86.5 Å^3^, the number of holes significantly increased with higher RAF content, which could explain a greater overall free volume available for the diffusion of gas molecules after a certain level of crystallinity is reached in PLA samples. More recently, these results have been further investigated and clarified by other authors [61,87]. For example, Fernandes et al. [61] prepared a set of PLA samples with well-defined microstructures, spherulite size, and RAF content as a function of crystallization temperature and annealing time. It was observed that the extent of the formation of RAF in the samples was the preponderant factor governing the oxygen permeability as the solubility and diffusion coefficients were seen to increase as a function of higher RAF content. Moreover, it was interesting to note that samples that were highly nucleated prior to crystallization provided the best results in terms of oxygen barrier properties, indicating that a pre-nucleation step and the short crystallization times hinder the formation of RAF. To close the loop, Sangroniz et al. [87], via a combination of techniques (i.e., PALS, DSC, and density tester), clearly confirmed that the formation of de-densified RAF in annealed PLA samples increased the free volume as the polymer chains had a more rigid conformation than in MAF. Moreover, they found that due to the higher solubility of RAF in water, the overall free volume was further increased due to the plasticization effect of the water molecules, and therefore both factors would contribute together to the increase in the permeability values. Overall, all of these studies reported thus far indicate that an increasingly higher amount of RAF has a detrimental effect on the PLA barrier properties. To minimize the conversion of MAF into RAF with time, a pre-nucleation step and short crystallization time at high temperature may be used.

### 3.3. Polymer Drawing

Another common way to improve the polymeric materials’ barrier properties is through molecular orientation, whereby a polymer is stretched in either one (uniaxial) or two (biaxial) predetermined directions (i.e., by extrusion or injection molding). In other words, orientation involves the use of mechanical and thermal energy to rearrange the polymer in an oriented position at the molecular level. The reduced gas and vapor permeation of oriented semicrystalline polymers is well documented [66,86,93,94,95,96], and the main mechanism involves the transformation from a spherulitic structure (the lamellar crystals propagate radially from the nucleation site) to a densely packed microfibrillar conformation (growth of lamellar crystals perpendicular to the direction of strain). This transformation induces an effective reduction in the diffusion coefficients by increasing the tortuosity of gas transport paths [66]. In the specific case of PLA, while the use of molecular orientation has been widely implemented for improving its thermal and mechanical properties, a limited number of papers could be found on how the drawing process affects its permeability behavior [62,97,98,99]. One relevant example is the study conducted by Delpouve and co-workers [62], whereby the effects of three different drawing modes on the water permeability properties were investigated on compression-molded PLA films (Table 3). Samples were subjected to uniaxial constant width (UCW) drawing (films are drawn only in one longitudinal direction (LD), simultaneous biaxial (SB) drawing (films are drawn in two perpendicular directions), and sequential biaxial (SEQ) drawing (films are first drawn in LD then in the transversal direction). The permeability behavior of the drawn samples was compared to those of fully amorphous and thermally unoriented crystallized samples. As shown in Table 3, results showed that the water permeability coefficient (P) of the drawn materials were in all cases lower than the amorphous and the thermally unoriented crystallized samples, regardless of the drawing modes used. Among all of the drawn samples, a significant decrease in the permeability coefficient was obtained with the SB drawn samples, followed by the UCW and SEQ ones. In particular, the water permeability was observed to decrease from 2.15 ± 0.07 × 10^−12^ for the amorphous film (maximum *p* value) to 1.63 ± 0.07 × 10^−12^ mol m^−1^ s^−1^ Pa^−1^ for the SB sample (minimum *p* value), accounting for about 35% of reduction.

As shown in Figure 4 (WAXD patterns), the best performance of SB compared to those obtained via uniaxial or sequential drawing modes was ascribed to the resulting homogeneous orthotropic structures of the SB drawn films, whereby PLA macromolecules were oriented perpendicularly to the water diffusion path. Between UCW and SEQ, the resulting worse performance of SEQ compared to UCW samples was linked to partial destruction of the crystallites because of the two sequential chain orientations. Likewise, Dong et al. [99] investigated both the oxygen and water vapor permeability of two types of uniaxial drawn PLA samples, one simply stretched with a twin-screw extruder system and the other one was first stretched and then annealed at 90 °C for two hours. All samples were stretched at different draw ratios, namely R = 1, 2, 3.5, 5, and 6.5. While the sample crystallinity content was seen to increase as a function of draw ratio, the morphological analysis of the samples’ surfaces indicated that too high a stretching strength (higher than R = 3.5) promoted the formation of cracks and a porous structure with high permeability to both gas and water vapor. However, samples with draw ratios up to 3.5 showed smooth surfaces. Compared to the undrawn PLA films, both the stretched and annealed samples (at R = 3.5) showed a significant reduction (~25%) in the oxygen and water vapor permeability. A slightly better performance was obtained with the annealed samples, and this was attributed to the annealing process, which increased the density of the annealed PLLA films relative to the simply stretched one. These results confirm that the drawing process does promote higher barrier properties in PLA films compared to the corresponding undrawn samples. However, care must be taken in order not to exceed the maximum tolerated stretch, as this could have a negative impact on both film morphology and permeability performance.

### 3.4. Nucleating Agents

As widely discussed in the previous paragraphs, the crystallization process and the resulting crystal size, orientation, and morphology can have a drastic effect on a wide range of polymer physical properties including the gas and water vapor permeability. PLA crystallization has been shown to be very slow [100,101,102], making common polymer processing operations (i.e., injection molding, extrusion, fiber spinning, melt blowing, etc.), time-consuming steps for industrial production [103]. One way to speed up these processes involves the use of effective nucleating agents, which will lower the surface free energy barrier toward nucleation and thus promote faster crystallization rates. Moreover, depending on the type, size, and aspect ratio of nucleant particles, preferential crystal orientation and/or specific crystal superstructures with tailored properties can be obtained [104,105,106,107,108]. Typical nucleating agents for PLA include talc, lactide, montmorillonite, boron nitride, calcium carbonate, magnesium carbonate, titanium oxide, or graphene oxide, to name a few. While these have all shown to significantly increase PLA crystallization rate to a greater or lesser extent [102,109,110,111], only a few of them have also been found to be particularly effective in improving its barrier properties through different mechanisms. For example, Ghassemi et al. [105] demonstrated that the addition of 3 wt% of talc resulted in a 25–30% decrease in the gas permeability and diffusion coefficients of extruded PLA films to five different gases (hydrogen, oxygen, dioxide carbon, nitrogen, and methane). This was ascribed to the plate-like structure of talc that limited gas motion within the PLA matrix (increase in the tortuosity). Likewise, Buzarovska et al. [112] demonstrated that the addition of 5 wt% of talc lowered the water vapor permeability from 6.71 × 10^−12^ (neat PLA) to 2.96 × 10^−12^ mol m/m^2^ sPa in solution-casted PLA films, resulting in up to 55% decrease in the overall water permeability compared to neat PLA. Additionally, in this case, the resulted lower permeability was ascribed to an increase in the diffusion distance and tortuous path for the permeants and this was correlated to an even distribution of impermeable platelet talc particles within the PLA matrix. Another relevant example to tailor PLA properties is through the stereocomplex (SC) crystallite formation between enantiomeric PLLA and PDLA, which can be prepared by a simple physical blending route. As shown in Figure 5A,B, depending on the ratios of PLLA/PDLA, the resulting mix acts as a proper nucleating agent by promoting simultaneous folding of the two enantiomeric chains in triangular (for non-equimolar blends) or hexagonal (equimolar mix) shapes [113,114,115,116].

These peculiar conformations offer favorable positions for the polymer loops during the crystal growth, which in turn, leads to a significant reduction in the induction period for the formation of highly dense spherulites. Given the advantageous crystallization process, the potential effects of stereocomplexation on PLA barrier properties were also investigated [117,118,119,120,121]. For example, Tsuji and Tsuruno [117] found that the water vapor permeability values of PLLA/PDLA solution-casted stereocomplex-based films (SC-PLA) were significantly reduced compared to those of the corresponding homopolymers. In particular, the WVP of SC-PLA was reduced by 14–23% than those of pure PLLA and PDLA with $X_{c}$ in the range of 0–30%, indicating that superior barrier properties were achieved, even for fully amorphous SC-PLA. In a more detailed study, Varol et al. [121] conducted an in-depth investigation into the effect of PLA stereocomplexation on the transport properties toward a wider range of permeants (water, nitrogen, oxygen, and carbon dioxide). The water permeation data revealed a drastic barrier improvement of up to 70% for SC-PLA with respect to the corresponding homopolymers with the same $X_{c}$ content. Moreover, it was interesting to note that the water permeability coefficient of fully amorphous SD-PLA at 25 °C was comparable with those of pure semicrystalline PLLA and PDLA ($X_{c}$ = 48%). More interestingly, the gas barrier properties of SC-PLA toward all gases were exceptionally enhanced compared to both the parent homopolymers. For example, the permeability coefficient of pure PDLA and PLLA was seen to decrease from 339 ± 44 and 71 ± 25, respectively, up to 0.14 ± 0.02 (Barrer) in SC-PLA measured at the same conditions. Similar trends were observed toward N_2_ and CO_2_. The overall significant improvement in the barrier properties of SC-PLA was attributed to a change in the crystal conformation from the common α type (pseudo-orthorhombic unit cell with 10_3_-helical chain conformation) of the pure homopolymers to the triangular-like crystal shape (triclinic unit cell with 3_1_-helical chain conformation) for SC-PLA, which resulted in a smooth and non-porous polymeric matrix. These results clearly highlight the importance of the crystal superstructure on the barrier performance of semicrystalline PLA. In line with this research direction, many authors have reported that the addition of specific nucleating agents could manipulate the crystal superstructure of polymers [103,122,123,124]. In particular, it was recently shown that 1,3,5-benzenetricarboxylamide derivatives (known family of amide nucleating agents for polypropylene) can tailor the crystal superstructure of PLA, affording three distinct crystal morphologies by melt crystallization such as cone-like, shish-kebab, and needle-like structures [103]. As demonstrated by in situ polarized optical microscopy (POM) and rheological measurements, the nucleant easily dissolves in PLA melt and can self-organize into fine fibrils prior to PLA crystallization. These fibrils act as shish, from which the peculiar PLA crystal structures are formed, depending on the amount of nucleant added (0–0.5 wt%). Among all structures, the shish-kebab-like structure obtained at 0.3–0.5 wt% was further explored (by the same author) as a potential ideal conformation to enhance the barrier properties of PLA [125]. In fact, as shown in Figure 6A,B, the epitaxial growth of crystals occurred orthogonally to the long axis, which, in turn, could form a densely packed wall structure along a vertical direction of the gas diffusion path. The oxygen permeability results shown in Figure 6C display a drastic improvement in the barrier properties of PLA crystallized with increasingly higher amounts of N,N′,N″-tricyclohexyl-1,3,5-benzene-tricarboxylamide (trade name TMC-328), an active model nucleating agent belonging to the 1,3,5-benzenetricarboxylamide derivative family. The best oxygen permeability value was observed for the PLA sheet containing 0.5 wt% of nucleant (*p* = 1.989 × 10^−20^ m^3^·m/m^2^ sPa), exhibiting a reduction of about 300× compared to the corresponding parent PLA sheet with isotopic spherulitic crystals (*p* = 5.244 × 10^−18^ m^3^ m/m^2^ sPa). In a similar study, comparable results of enhanced barrier properties for PLA due to a shish-kebab-like structure were obtained using a different active nucleating agent belonging to the benzhydrazide family, namely octamethylene dicarboxylic dibenzoyl hydrazide (TMC-300) [126]. Like TMC-328, TMC-300 fibrils induced epitaxial growth of the PLA lamellae orthogonally to their fibrillary direction, affording “lamellae-barrier walls” stacked perpendicular to the direction of gas diffusion. By employing a series of layer-multiplying elements at the end of the extrusion setup, multilayer samples with up to 64 layers were produced. The oxygen permeability coefficient showed a gradual reduction with increasing layer number of the dense and impermeable “lamellae-barrier walls”, reaching the lowest value (up to 85.4% decrease) with the 16-layer sample compared to the control one. This particularly high resistance to gas permeation was associated with the highest content of branch fibrils in the 16-layer sample that contributed to a more regular in-plane arrangement of PLA lamellae. The impact of multilayer formation on the barrier properties of PLA-based films is further discussed in the following section.

### 3.5. Nanoconfinement Approach

Another way to improve the performance of packaging materials is based on the formation of coatings or multilayers [127,128]. In this context, the use of nanotechnology has recently shown that the formation of nanoscale layers (i.e., nanolayers) can result in unique crystalline morphologies that can have a profound impact on the barrier properties of the packaging films [129]. Common technologies for developing these nanostructured multilayer films include the layer-by-layer (LbL) technique [130], the electrohydrodynamic processing (EHDP) [131], and the layer-multiplying “forced assembly” coextrusion (LmFAC) method [132]. The latter method, introduced 40 years ago by Dow and recently updated by Baer’s group, has attracted particular attention because it can control the crystallization habits of polymers and improve properties not possible with the bulk [129]. Briefly, this technique consists in combining two or three polymers into a continuous alternating layered structure with hundreds or thousands of layers of nanometric thickness. The molecular and chain organization of the polymers in a confined nanometer-scale space (ultrathin films) prevents isotropic spherulitic growth of the lamellar crystals, resulting in unique crystal orientations. As can be seen in Figure 7, these are “on-edge”, where the polymer chains are oriented parallel to the substrate plane, and “in-plane”, where the polymer chains are oriented perpendicular to the substrate plane [133]. Due to the high-aspect ratio and peculiar orientation of the highly ordered interlayer lamellae, the latter has led to a substantial reduction in the gas permeability by several orders of magnitude compared to their bulk film controls for a wide range of confined/confiner polymers [118,134,135,136] including PLA, albeit in a very limited part [137,138,139]. The general principles of mass transport in polymeric multilayers have been carefully summarized elsewhere, so we refer readers to the following articles: [140,141].

The orientation of crystals is undoubtedly a crucial parameter for tailoring barrier properties and is directly affected by crystallization temperature, film thickness, chain mobility, and substrate–polymer interactions. However, when PLA was chosen as a model semicrystalline polymer to study the impact of confinement on barrier properties, the results showed that the dynamics of the amorphous phase (i.e., the occurrence of RAF in the multilayer samples) also plays a crucial role in the overall barrier performance [133,134,135,136,137].

For example, Messin et al. [137] provided new insights into the relationships between microstructure implying RAF and barrier performances of a 2049-layer film of poly(butylene succinate-co-butylene adipate) (PBSA) confined against PLA prepared using LmFAC technology. The content of the multilayer was 80 wt% PLA and 20 wt% PBSA. The continuity of the PLA/PBSA layers can be clearly seen in Figure 8A using an atomic force microscopy (AFM) image. The confinement effect caused by the PLA resulted in a slight orientation of the crystals in both the transverse and extrusion views and an increase in RAF in PBSA with densification of this fraction. As shown in Figure 8B, these structural changes significantly improved the water vapor and gas barrier properties of the PBSA layer by up to two decades in the case of CO_2_ gas, mainly due to the reduction in solubility. However, it is important to note that the results of these authors contradict recent findings from other studies (see Section 3.2) that RAF is responsible for a de-densification of the amorphous phase and a decrease in the overall barrier performance. In this context, Nassar et al., in a more recent study [138], investigated the barrier properties of PLA in multinanolayer systems with two amorphous polymers (polystyrene, PS; and polycarbonate, PC), probing the effect of confinement, the compatibility between the confiner and the confined polymer, crystal orientation, and amorphous phase properties. WAXD measurements showed that the PLLA lamellae between PS layers had a mixed in-plane and on-edge orientation, while the PLLA lamellae between PC layers were clearly oriented in-plane. More importantly, the RAF content of semicrystalline PLLA was about 15% in PC/PLLA, whereas it was negligible in PS/PLLA. Oxygen permeability results showed that the occurrence of RAF in PC/PLLA samples had a detrimental impact on the barrier properties of the multilayer films, which could not be compensated by the presence of in-plane crystals. Moreover, annealing PS/PLLA films to minimize RAF content allowed for a barrier improvement of the PLLA layers by a factor of two compared with semicrystalline bulk PLLA. Overall, it can be pointed out that work conducted thus far in this regard is still limited and contradictory. Therefore, further investigation is recommended to elucidate the effects of amorphous-phase dynamics and nanoconfinement on the barrier properties of PLA-based multilayer films.

## 4. Bio-Blends and Composites

Blending is a well-established low-cost technology to develop next generation plastics with enhanced properties compared to the single components. Polymer blending is indeed a straightforward and user-friendly process, whereby materials are easily and rapidly mixed either in co-solution or in the molten state. It is therefore not surprising that this approach has been largely applied for improving PLA performance in many applications [142,143,144] including food packaging [12,145,146,147]. Among the uncountable number of co-additives employed, bio-fillers and other biodegradable polymers have recently attracted a great deal of attention, viewed in the context of minimizing environmental impacts and encouraging greater use of sustainable and renewable sources [148,149,150,151,152,153]. Those materials include natural fibers, especially as reinforcing agents (i.e., wood, cellulose, sisal, kenaf, flax, and hemp) and common biodegradable polymers (lignin, starch, polycaprolactone (PCL), polybutylene succinate (PBS), and polyhydroxyalkanoates (PHAs)). Moreover, the implementation of nano-additives (i.e., nanoclay, nanowhiskers, and 3D-isodimensional nanoparticles) for the formation of bionanocomposites in the field of food packaging have rapidly increased over the last decade due to their exceptional ability to improve PLA film properties [154,155,156,157,158,159]. However, as for all blends and composites (polymer–filler, polymer–polymer, polymer–nanomaterial), the resulting properties may vary widely as they do not depend only on the intrinsic nature of the components, but they are also highly affected by the final morphology. As shown in Figure 9, common morphologies of blends include laminar, ordered, fibrillar, droplet-type, and co-continuous. Each structure has its own advantages in terms of mechanical, thermal, and barrier properties. However, among all, laminar and co-continuous microstructures are the most desired ones for barrier improvement as the two phases are complementary reversed and the surface of each phase is an exact topological replica of the other, both contributing equally to the blend properties [160]. To obtain these structures, besides the degree of miscibility between components, the volume fraction and the choice of appropriate compatibilizers play a fundamental role in the achievement of a high degree of dispersion, which in turn, determines the overall end-product quality [161,162,163,164]. In the following paragraphs, the most relevant PLA-based bio-blends and composites are briefly reviewed in terms of the morphology–property relationship to give an update on the progress made in the improvement of PLA barrier properties.

### 4.1. Bio-Based Reinforcing Agents

The use of bio-based reinforcing agents to produce PLA-based biocomposites with improved properties has become one of the key investment trends [165,166,167,168,169,170]. In this context, low-cost bio-renewable fibers such as cellulose, wood, kenaf, sisal, flax, and hemp have received increasing attention due to their reinforcing ability, non-toxicity, low-density, and large availability. As the name implies, these materials are commonly introduced into the polymer matrix as a reinforcement, thus enhancing its mechanical properties and stability. Concerning the barrier properties, little work has been conducted so far, as these materials are hydrophilic in nature with aa high tendency to adsorb water from the environment [171]. Moreover, the very low affinity with hydrophobic polymers (i.e., PLA) further hinders the formation of a well-dispersed system, which is one of the key factors for an efficient barrier. In light of the foregoing, Sanchez-Garcia et al. [172] investigated the morphology and transport properties of solvent-casted PLA biocomposites loaded with different ratios of purified alfa micro-cellulose (MC) fibers. Scanning electron microscope (SEM), AFM, and Raman imaging investigations showed that a good degree of dispersion was obtained only for samples with the lowest MC content (1 wt%), while at higher fiber incorporation (i.e., 10 wt%), clear presence of fiber agglomerates and phase discontinuity was reported to increase as a function of loading. As a matter of fact, water permeability data showed that the barrier properties of PLA biocomposites was only reduced by 10% in the sample containing 1 wt% of fiber content and at higher content, the permeability was seen to increase by 80%. Likewise, Luddee et al. [173] prepared a series of PLA biocomposite films containing grounded bacterial cellulose (BC) as a reinforcement and the water permeability behavior was studied as a function of filler particle size. Results showed that the incorporation of BC led to an increase in the water vapor permeability for all biocomposites compared to neat PLA. Moreover, the permeability coefficients increased linearly with the BC particle size, suggesting that BC particle sizes greatly affected the filler dispersability and their tendency to agglomerate within the PLA matrix, as also confirmed by SEM images. In order to improve compatibility between components, natural fillers (with free OH groups) can be chemically surface-modified with coupling agents. Those involve alkaline and peroxide treatment [174,175], vinyl grafting [176], acetylation [177], bleaching [170], and organosilane coating [178], to name a few. These types of treatments aim to increase the interfacial bonding strengths between the natural fibers and the polymeric matrix either through the formation of covalent bonds or mechanical interlocking. For instance, D. Kale et al. [179] carried out surface acylation of microcrystalline cellulose (MCC) to reduce the overall filler polarity. The resulted acylated MCC (AMCC) was loaded into the PLA matrix and the barrier performance of the resulting PLA biocomposites investigated. As expected, results showed that the addition of untreated MCC in the PLA matrix increased the water vapor permeability coefficients due to the low degree of dispersion and particle agglomeration compared to neat PLA. In contrast, the addition of AMCC resulted in a better filler-PLA dispersion and the overall water permeability was reduced by up to 10% compared to pure PLA. On the other hand, Tee et al. [180] investigated the barrier properties to water vapor and oxygen of the PLA biocomposite containing silane-grafted cellulose (SGC) as a filler. While water permeability coefficients increased for all biocomposites tested (with treated and untreated cellulose), oxygen permeability values decreased by up to 50% for biocomposites containing 30 wt% SGS compared to pure PLA. This was attributed to the improvement in interfacial adhesion between the filler and matrix and the higher degree of affinity after silane treatment.

### 4.2. Biodegradable Polymers Blends

Except for a few cases (i.e., PCL and PHAs), most of the biodegradable polymers that have been co-blended with PLA are highly polar in nature. Similar to the above case of hydrophilic reinforcing agents, blending immiscible polymers results in poor interfacial adhesion and phase-separated systems, which typically show very low overall performance; thus, appropriate compatibilization must be accomplished to obtain the desired end-properties. Among all compatibilization strategies, the use of reactive coupling agents and catalysts is considered the most viable solution for industrial application. As an example, there has been a great research interest in blending PLA with several types of starch (i.e., granular and thermoplastic) for many applications including food packaging due to its good food contact compatibility, suitable barrier properties, and low cost. However, the hydrophilic nature of starch leads to the formation of a two-phase system with very poor properties [181,182,183]. Many compatibilizers have been used and those include glycerol [184], polyethylene glycol [185], citric or stearic acids [186,187,188,189,190], maleic anhydride [187], lignocellulosic materials [188], methylenediphenyl diisocyanate [189], and adipate or citrate esters [190]. In most of the works conducted in this regard, thermoplastic starch (TPS) has been preferred over naturally granular starch as it can be deformed and dispersed to a much finer state, which in turn, greatly improves material processability. For instance, Shirai et al. [191] investigated the barrier properties of PLA/TPS biodegradable sheets with citric (CA) and adipic (AA) acids as additives (in the range of 0–1.5 wt%), prepared by a calendering–extrusion process at pilot scale. Prior to mixing, PLA and TPS were separately plasticized with 10 wt% of diisodecyl adipate (DIA) and 30 wt% of glycerol, respectively, as this pre-plasticization step has been reported to significantly improve the blend processability (extrusion). As shown in Figure 10A, all formulations could be processed continuously and the resulting sheets containing CA had a much smoother surface, homogeneous distribution, and compact structure compared to the other formulations, which, in contrast, revealed rough surfaces due to partial phase separation. The water vapor permeability studies showed that sheets containing CA exhibited the lowest WVP value, which accounted for about 70% of reduction compared to the plasticized reference sample. This was associated with a more effective interaction between starch and PLA (increases the interfacial adhesion), which in turn promoted mobility reduction over polymeric chains. The higher CA concentration (1.50 wt%) did not improve the evaluated property, suggesting that this component is efficient at lower concentrations (0.75 wt%). Comparatively, AA did not exhibit the same performance, even when mixed with CA, the WVP values were even increased in the presence of AA, probably due to partial degradation through acidolysis. In another interesting study, Muller et al. [192] studied the effect of cinnamaldehyde incorporation on the properties of the amorphous starch-PLA bilayer films intended for packaging applications. The particular compatibilizer was chosen as it is classified as GRAS (Generally Recognized As Safe) by the FDA (Food and Drug Administration) and offers antibacterial, antifungal, anti-inflammatory, and antioxidant activity to the resulting packages. The PLA/starch bilayer films were successfully obtained by compression molding followed by thermo-compression and the barrier performance studied in the presence and absence of cinnamaldehyde. Despite the lower ratio of the PLA sheet in the bilayer assembly (1/3 of the film thickness), a significant improvement in the gas barrier properties was achieved in the absence of the compatibilizer, specifically, a 96% decrease in WVP with respect to the neat starch films and a 99% decrease in OP compared to the amorphous PLA films. Surprisingly, when cinnamaldehyde was added, films exhibited lower mechanical resistance due to relevant plasticization of the amorphous regions, and WVP and OP did not notably change.

PHAs represent another relevant class of polymers that offer great potential in the food packaging industry due to good thermomechanical and barrier properties [193,194,195,196,197,198,199,200,201,202]. PHAs are derived from renewable resources and due to their bacterial origins, this class of polyesters shows good degradability features [197]. Poly(3-hydroxybutyrate) (PHB) is a homopolymer of 3-hydroxybutyrate and is the most common type of the PHA family, together with its copolymer with polyhydroxyvalerate (PHV), PHBV, which shows superior flexibility and processability. Due to the great potential of this class of polyesters, PLA-PHAs blends have attracted increasing interest in the last two decades [196,198,199,200]. Although both PLA and PHAs are compatible polyesters, several studies have shown that they form miscible blends only when low molecular weight (MW) fraction polymers are mixed and/or low polymer ratios (up to about 25 wt%) are incorporated into one another [21,193,201,202]. As an example, Boufarguine et al. [21] demonstrated that blending PLA with only 10 wt% PHBV using a multilayer co-extrusion process resulted in well dispersed films (up to 17 layers) and the gas barrier properties significantly improved compared with neat PLA. In particular, the helium permeability showed a reduction of about 35% with respect to pure PLA. In another study conducted by Zembouai et al. [193], it was shown that PHBV/PLA blends prepared by melt mixing in different ratios (100/0, 75/25, 50/50, 25/75, and 0/100 wt%) were not miscible, forming a two-phase system at all compositions (see Figure 10B). On the other hand, the water and oxygen barrier properties of PHBV/PLA blends were significantly improved with increasing PHBV content. At a PHBV content of 75 wt% in the blend, a reduction of about 75% and 81.5%, respectively, was achieved compared to pure PLA. However, even at a PHBV content of only 25 wt%, a reduction in the permeability coefficients for oxygen and water of about 35.3% and 22.7%, respectively, was obtained. This apparently anomalous behavior was further investigated in a comprehensive study reported by Jost and Kopitzky [202], whereby the miscibility of PLA-PHBV cast films and the resulting barrier properties were reviewed under thermodynamic aspects and correlated to their experimental results. In addition, blends were produced with different polymer molecular weight fractions and the final properties were studied in the presence and absence of selected compatibilizers. It was found that the incorporation of PHBV into the PLA matrix in the range between 20 and 35 wt% resulted in miscible blends. The reference blend (PLA:PHBV 75:25; MW: 10–40 kDa) with or without compatibilizers showed lower permeability values (to water vapor and oxygen) than the calculated values for the corresponding system. In agreement with previous findings (miscibility, crystallization, and melting of PHBV/PLA blends), this phenomenon was ascribed to the quick formation of interpenetrating PHBV spherulites, which interlock with the PLLA structures, leading to a reduction in the overall free volume. This may explain why, even for immiscible blends, the incorporation of PHBV into the PLA matrix enhances the barrier properties of PLA.

## 5. PLA-Based Nanocomposites

Nanocomposites are multicomponent systems, whereby the major constituent is typically a polymer and the minor one consists of a material with a length scale below 100 nm, referred to as nanofiller or nanoload. Nanoparticles can be classified into three major categories according to their particle geometry:(i).layered nanoparticles that are characterized by one dimension ranging from several angstroms to several nanometers (i.e., layered silicates);(ii).elongated particles that consist of fibrils with a diameter ranging between 1 and 100 nm and length up to several hundred nanometers (i.e., cellulose nanofibers); and(iii).isodimensional particles that have the same size in all directions and an aspect ratio close to unity (i.e., metal oxide nanoparticles).

Due to clear evidence of their outstanding performance in many applications [203,204,205], it is no wonder that these systems have attracted a great deal of research interest worldwide in recent years. In the packaging field, introducing impermeable nanofillers with high aspect ratio and large surface area in the polymer matrix has appeared to be a promising approach to enhance the barrier properties of polymers [206,207]. As shown in Figure 11, this can be achieved by two specific mechanisms [208]: first, a tortuous path for gas diffusion is created though the polymer matrix as the evenly dispersed nanofillers are impermeable inorganic particles and act as an obstacle; and second, the nanoparticulate fillers may positively interact with the polymer matrix, “immobilizing” the polymer strands at the polymer–nanofiller interface and thus decreasing the overall free volume available for the gas molecules to diffuse through the polymer surface. It is therefore clear that nanocomposites offer encouraging opportunities for the food packaging industry. In the following paragraphs, the progress made in the development of PLA-based nanocomposites as high barrier materials are briefly reviewed.

### 5.1. Layered Nanofillers

Among all available nanosystems, nanoplatelets of layered silicate clay are by far the most researched nanofillers [209,210,211,212]. As shown in Figure 12, layered silicates consist of very thin films associated with counterions (exchangeable cations) [212]. Based on the types and relative content of the unit crystal lamellae, they can form three different structures:(i).1:1 clay types: the unit lamellar crystal consists of only one crystal sheet of silica tetrahedra in combination with an octahedral sheet, i.e., single crystal lamellae of alumina octahedra;(ii).2:1 clay types: consist of two crystal layers of silica tetrahedra forming the unit lamellar crystal bounded by a crystal layer of alumina octahedra located in the middle of the two layers; and(iii).2:2 clay types: consist of four crystal layers, alternating crystal layers of silica tetrahedra and alumina (or magnesium) octahedra.

Typically, the layer thickness does not exceed 1 nm and the adjacent dimensions can vary from 300 Å up to several micrometers, depending mainly on the clay types and preparation methods [213]. For this reason, the aspect ratio (values > 1000) and the surface area (~700–900 m^2^/g) are particularly high. These particular morphological features usually lead to impressive improvements in barrier properties when the nanoclays are uniformly distributed in the polymer matrices. Several studies have focused on the preparation of PLA-based layered silicate nanocomposites with improved barrier properties [148,214,215,216,217]. Among all types, the impermeable 2:1 layered phyllosilicate montmorillonite (MMT) is certainly the most commonly used prototype clay for this application [216,217]. The particular layered structure of MMT, consisting of an octahedral layer (mainly composed of Al_4_(OH)_12_) intercalated between two tetrahedral layers (composed of SiO_4_ units), allows for the formation of multilayered polymer arrays with a high barrier. However, MMT, similar to most other clays, is inherently hydrophilic and has a high surface energy. This leads to a high segregation tendency and agglomeration of clay nanoplatelets, especially when dispersed in non-polar polymer materials (e.g., PLA) [218,219]. Agglomeration of clay platelets leads to the formation of tactoid structures with lower aspect ratios and thus lower barrier properties. To circumvent this problem, nanoclay surfaces are organically modified with cationic surfactants (usually quaternary alkylammonium compounds) by an ion exchange process with the inorganic cations naturally present in clay minerals [220,221,222]. Such a process reduces the surface energy of the silicates and the intercalated cationic surfactants act as compatibilizers between the hydrophilic clay and the hydrophobic polymer. The most common commercially available organically modified MMT clays include Cloisite (CH) 20A (with dimethyl dihydrogenated tallow ammonium chloride) and 30B (with methyl tallow bis-2-hydroxyethyl ammonium chloride), which have been shown to provide the greatest interlayer spacing and improved interactions with nonpolar polymers [223,224,225]. Moreover, to ensure homogeneous dispersion and delamination of the nanoclays in the polymer matrix, appropriate processing conditions should be applied. These may include, but are not limited to, high-pressure homogenization [226], a pre-sonication step [227] to disaggregate the clay platelets, and/or two-stage extrusion masterbatch processing to obtain a fine dispersion. The barrier performance of PLA-based nanocomposites containing CH has been studied by several authors [228,229,230,231,232,233]. Figure 13A,B shows two examples of successful CH 30B/PLA-based nanocomposites in terms of barrier performance [231,234].

Other examples include the work of Najafi et al. [233], in which the effects of different processing conditions on the dispersion of 2 wt% CH 30B in PLA-based nanocomposites were studied and the resulting film morphologies were subjected to the oxygen permeability test. PLA–clay mixtures were prepared with and without the chain extender Joncryl^®^ to further improve blend compatibility. Nanocomposites were prepared using a twin screw extruder with different methods. The preparation methods consisted of either simultaneous extrusion of all components together or a two-step extrusion masterbatch approach, with the chain extender added in either the first or second pass. In addition, the effect of multiple extrusion passes was also examined. According to the morphological observations conducted by SEM, TEM, and XRD, the best clay–PLA dispersions were obtained in the presence of Joncryl^®^ when processed in the extrusion masterbatch approach, while multiple extrusion passes led to the formation of large clay aggregates due to the longer extrusion residence time. As expected, the good dispersion and distribution of clay platelets in PLA–Joncryl-based nanocomposites resulted in the lowest measured oxygen permeability, which accounted for 37% of the reduction compared to pure PLA. However, it was interesting to note that simple addition of the chain extender into pure PLA increased the oxygen permeability of the corresponding blend. This was explained by the formation of long chain branches, which reduced the crystallinity of pure PLA from ~7% to ~1%. In another study, Tenn et al. [230] investigated the incorporation of various concentrations (from 0 to 20 wt%) of CH 30B (both in the hydrated and pre-dried states) into PLA films by two-step extrusion masterbatch processing. The transport properties (water and oxygen permeability) of the resulting systems were correlated with the degree of dispersion and orientation of the nanoplatelets in the polymer matrix. To investigate the effect of the hydration state of the clays on the quality of the dispersion, CH 30B was added both in partially hydrated form (as received) and in a pre-dried state (80 °C in a vacuum oven for 12 h). For both systems of PLA–CH (hydrated and anhydrous), a decrease in water and oxygen permeability values was observed as a function of increasing nanoclay content. This is generally due to the well-known tortuosity effect, which improves the diffusion path of the permeants in the PLA matrix. However, comparing the two nanocomposites at the same nanoclay content, it was found that the system containing the hydrated form of CH had a higher barrier effect than the corresponding systems containing the dried clay form (CD). In terms of water permeability (Pw), PLA films containing 15 wt% CH showed the best performance, achieving a 95% reduction in Pw compared to the pure PLA film. The difference in water barrier performance between the two systems was ascribed to the presence of water molecules in the untreated clay component, which favored the formation of water clusters that hindered water diffusivity. Additionally, in terms of oxygen permeability, the best performance was obtained with PLA–CH systems, which achieved a reduction of up to 74% compared to the PLA-only film. The overall better performance of the PLA–CH nanocomposites was further investigated in terms of dispersion quality and it was found that in all systems, there was a coexistence of intercalated and aggregated structures depending on the clay content. However, when CH was incorporated, the TEM images showed relatively higher intercalation of clay platelets, likely due to better compatibility between CH and PLA, as observed by DSC and XRD. In addition, CH nanoplatelets were found to preferentially arrange perpendicular to the diffusion pathway, confirming the fundamental role of structural orientation in the permeation behavior of materials. These and similar studies have shown that the uniform distribution of organically modified MMT layers in the PLA matrix is the key factor for significantly improving the barrier properties of PLA.

### 5.2. Nanofibers or Whiskers

With the raising environmental awareness and concerns over sustainability and end-of-life disposal challenges, the interest in exploiting nanomaterials from renewable resources is rapidly increasing [235,236]. In the packaging field, nanofibers derived from natural sources have attracted a great deal of attention, not only for their environmental friendliness over traditional nanofillers, but also due to their outstanding ability to decrease the permeability of various polymeric films [237,238,239,240]. These include all the nanofibrous materials derived from cellulose, starch, chitin, and chitosan, which commonly exist in plants, animals, microorganisms, and bacteria. Among them, cellulose-based nanofillers have been more widely investigated due to advances in the production of cellulose nanofibrils (CNF) and cellulose nanowhiskers (CNW) [241,242,243,244]. CNF have elongated rod-shaped structures with large diameters and lengths variations, both ranging from a few up to 100 nm. Due to their highly crystalline nature, large surface area/aspect ratio and their ability to form a dense percolating network, CNF are known to have high barrier properties toward most gases and liquids. However, since cellulose-based materials have water-loving surfaces due to the abundance of –OH groups, blending CNF (as it is) with hydrophobic materials such as PLA is not feasible without either the addition of compatibilizing agents or appropriate chemical modifications. Several strategies have been attempted to overcome this problem such as the use of surfactants [243,245], surface acetylation [246], and reactive compatibilization [247], to name a few. For example, Espino-Pérez and co-workers [243], used an in situ surface grafting method to produce fully compatibilized PLA/CNW bionanocomposites with enhanced barrier properties. The CNW surface was grafted with a long aliphatic isocyanate chain (n-octadecyl isocyanate (ICN)) and the barrier performance of PLA/CNW and PLA/CNW–ICN nanocomposite solution-casted films were evaluated. Visual observations of the films indicated that good filler dispersion and film transparency were obtained only with the lowest CNW and CNW–ICN content (2.5 wt%). At higher CNW concentration, aggregation of CNW in the matrix could be observed, indicating poor system compatibility. In fact, the water and oxygen permeability of both samples were not significantly reduced compared to neat PLA, likely due to the reduction in hydrogen bonds between the fibrils, which led to poor interfacial adhesion with the PLA matrix. This may suggest that while surface grafting of CNW with isocyanate is effective in enhancing the hydrophobicity of CNW, it is probably not the best compatibilizing approach to produce well-dispersed PLA–CNW systems. More recently, successful compatibilization and enhanced barrier properties of PLA–CNF nanocomposites was achieved by Nair and co-workers [248] by preparing CNF with high amounts of lignin (about 20–23 wt%) (NCFHL) from the bark of various coniferous species. Since lignin contains both polar (hydroxyl) groups and nonpolar hydrocarbon and benzene rings, its presence naturally enhances the hydrophobic nature and the barrier properties of CNF without additional modifications. As shown in Figure 14A–C, morphological examination of solution-casted PLA-NCFHL films with various NCFHL contents (5, 10, 15, and 20 wt%) revealed that with up to 10 wt% of load, fibrils were well-embedded within the PLA matrix.

As expected, the good dispersion observed at 5 and 10 wt% of NCFHL loading resulted in a significant enhancement in the water permeability performance of the nanocomposites, with WVTR values reduced to about half compared to neat PLA (Figure 14D) [248]. This was ascribed to the formation of an effective interphase between the lignin and PLA, which increased the tortuous path for water vapor diffusion. In this specific context, Rigotti et al. [249] carried out an in depth-investigation on the role of microstructure of PLA layers located at the clay–matrix interface on the nanocomposites’ gas transport properties. The barrier performance of solution-casted PLA films containing lauryl-functionalized cellulose nanofillers (LCNF) was examined using gas phase permeation measurements (toward CO_2_, H_2_, and D_2_ gases) and thermal desorption spectroscopy (TDS) analysis. Results showed that there exists a critical LCNF concentration (6.5 wt%) under which the permeation flux (P) for all gases decreases with increasing filler content, while at higher concentrations, flux increases, reaching a similar *p* value of neat PLA at 10 wt% of filler load. According to the SEM and TEM analyses, when filler content was lower than 6.5 wt%, LCNF appeared to be well-dispersed, while at higher concentrations, LCNF clusters and micrometric agglomerates could be observed. This is typically linked to a change in the nanostructure as a function of filler content. By analyzing experimental transport data using the permeation model of mixed matrix membranes, it was possible to conclude that at concentrations not exceeding 6.5 wt%, the interfacial regions at the filler–matrix interface were rigidified, likely due to strong lauryl–PLA chain interactions, which led to local free volume reduction. Therefore, this was suggested to be responsible to the improved barrier performance of PLA–LCNF systems, confirming that appropriate control of the nanocomposite interface properties is necessary to obtain systems with enhanced barrier capabilities.

### 5.3. Isodimensional Nanoparticles

Nanoparticles with three nanodimensions (less than 100 nm), also known as 3D-isodimensional nanoparticles, have gained increasing interest in the field of food packaging [250,251]. These include nanoparticles derived from most metals such as silica [252], copper [253], gold [254], silver [255], zinc [256], magnesium [257], titanium [258], and their corresponding oxides [259]. Among them, metal oxides have attracted special attention because they can be produced much more cheaply and are known for their strong antibacterial and ethylene scavenging activity [260]. These particular properties have shown that they offer an intriguing potential for the development of active nanocomposites for the packaging of fresh products, which are very sensitive to microbial spoilage. Several multifunctional PLA-based metal oxide nanocomposites have been prepared and the resulting transport properties investigated [261,262]. In a study by Lizundia et al. [261], the effect of incorporating different metal oxide nanoparticles (TiO_2_, SiO_2_, Fe_2_O_3_, and Al_2_O_3_) at 1 wt% on the transport properties of PLA films was investigated. TEM analysis showed that all nanoparticles had a spherical shape with similar diameters ranging from 10 to 20 nm. However, for the nanocomposites, it was interesting to note that SiO_2_ and Al_2_O_3_ isotropically distributed in the PLA matrix, while TiO_2_ and Fe_2_O_3_ formed large aggregates of about 100–150 nm. Unexpectedly, the most efficient metal oxide in the water permeability measurements was TiO_2_, whose incorporation resulted in a reduction of about 18% compared to pure PLA, followed by Al_2_O_3_ and Fe_2_O_3_ particles, which also resulted in a relatively moderate reduction (13% and 16%, respectively). The least efficient system was PLA–SiO_2_, which caused only 4% of reduction in water permeability. A different trend was observed with respect to oxygen: SiO_2_ and TiO_2_ improved the barrier performance of PLA to oxygen, while Al_2_O_3_ and Fe_2_O_3_ increased the overall permeability to about 5%. The author concluded that in these particular cases, the morphological characteristics (such as size, filler dispersion, and free volume) did not seem to play the most important role in the permeation process. The different barrier behavior observed in the samples was related to differences in the nature of the nanoparticles and possible filler–matrix interactions. In both cases, the best performance of TiO_2_ was attributed to its large hydrophobicity, while the worst barrier performance of SiO_2_ for water molecules was related to its high hydrophilicity, which favored a faster diffusion path. Moreover, PLA–ZnO antibacterial nanocomposite films were prepared in a composition range from 0.5 to 3 wt% of nanofillers by melt extrusion, and the barrier properties to water vapor were analyzed by Pantani et al. [262]. Since untreated ZnO can lead to severe degradation of the PLA matrix during the melt mixing process due to transesterification and depolymerization reactions, the surfaces of the nanoparticles were previously treated with protective agents (silane), which also acted as compatibilizers. TEM images in Figure 15A show a relatively continuous and well-dispersed rod-like ZnO distribution in the PLA matrix for all compositions [262]. However, only a slight reduction (~20%) in water vapor permeability (6.43 × 10^−7^ wt%/atm × cm^2^/s) was obtained with 3 wt% of filler compared to unfilled PLA (8.26 × 10^−7^), which was attributed to an increased difficulty for molecules to diffuse into the polymer matrix. In a similar work, Shankar et al. [158] prepared solution-cast PLA–ZnO nanocomposite films with concentrations of less than 2 wt% (0.5, 1.0, and 1.5). Morphological analysis reported in Figure 15B shows that the ZnO nanoparticles had a cubic and rod shape with a size in the range of 50–100 nm. However, it is worth noting that the surface of the nanocomposite films was relatively rough compared to that of neat PLA and the overall roughness appeared to increase with increasing ZnO content. Nevertheless, a reduction in the water vapor permeability of up to 30% was achieved compared to unfilled PLA, even for the composite with the lowest ZnO content (0.5 wt%). No further reduction was observed at higher ZnO contents, likely due to the formation of ZnO-based microagglomerated structures. In another study, Marra et al. [263] reported that homogeneous dispersion of 1 wt% ZnO in PLA-based nanocomposites in an increase in water vapor transmission rate to about 16% compared to normal PLA. This was associated with potential changes induced by the presence of ZnO particles at PLA–filler interfaces, resulting in an increase in free volume. In the same way, Swaroop and Shukla [264] studied the barrier properties of solution-cast PLA-based films reinforced with MgO nanoparticles. At 1 wt% metal oxide content, the resulting films showed a 20% increase in water vapor permeability. Morphological studies revealed that the nanoparticles were in the form of agglomerates, resulting in a very rough film surface. This was probably the main cause of changes in free volume, absorption, and solubility at the interfaces near the “highly polar” MgO nanoparticles, which overall contributed to the high water permeability measured. These results suggest that spherical composites with metal oxide exhibit relatively poor filler–polymer compatibility because the polarity of the particles does not match that of the PLA matrix.

## 6. Other Emerging Approaches toward High-Barrier PLA-Based Plastic

With the advent of new polymerization techniques, PLA-based block and graft copolymers with tailored properties have been synthesized based on judicious selection of co-monomers and the variation of copolymer compositions [265,266,267]. However, due to the good biocompatibility of PLA, the great majority of these copolymers have been mostly exclusively studied for applications in the field of biomedicine [268,269,270,271,272] and only little attention has been paid to the design of “high barrier” PLA-based copolymers for the food-packaging sector [17,273,274,275,276,277,278]. A few random studies, mainly focusing on improving the mechanical properties of PLA by copolymerization, have also shown somewhat promising results in terms of gas permeability performance of the resulting copolymers. For example, Genovese et al. [273] synthesized PLA-based ABA triblock copolymers with a hydroxyl-terminated poly(propylene/neopentyl glycol succinate) copolymer as a midblock, through ROP followed by chain extension reaction. Depending on A/B ratio, the resulting bio-based copolymers showed enhanced mechanical and barrier properties compared to the PLA homopolymers. In particular, the oxygen permeability of the copolymer containing 33% of the midblock unit was two times lower than that of neat PLA. These results were directly related to an increase in the degree of crystallinity and the presence of the two methyl groups in the neopentyl glycol sub-unit, which was suggested to hinder the passage of small molecules. Other examples include the copolymerization of PLA with rubbery-type monomers to afford versatile thermoplastic elastomers (TPEs) with good barrier properties [277]. TPEs are commonly referred to as ABA triblock copolymers containing an incompatible A hard block (component with high Tg) and a B soft block (component with low Tg). Typical examples include petroleum-derived styrene-based TPEs (PS–TPEs), which are extensively used in the packaging field [278] due to their well-known versatile properties and low cost. In fact, depending on the ratio between the soft and hard components, these materials’ properties can be easily tuned based on the required applications, ranging from slightly flexible plastics to highly elastic gums. With the increasing demand for eco-friendly alternatives, PLA-based TPEs could potentially substitute styrenic-based TPEs in this specific field. For example, Ali et al. [272] synthesized four different PLA thermoplastic polyurethane (PLAPU) copolymers with different compositions of hard PLA and soft PCL units via a chain-extension reaction. Depending on PCL content, the copolymers exhibited excellent flexibility and gas barrier properties. In particular, at the highest PCL content, the oxygen permeability was reduced by approximately 85 times compared to that of the PLA homopolymer. This behavior was ascribed to the incorporation of high molecular weight PCL segments, which led to an increase in the chain density, thus decreasing the effective path for diffusion. In a more recent study, Yuk et al. [274] prepared a series of thermoplastic superelastomers based on poly(isobutylene)-graft-poly(l-lactide) copolymers by a “grafting from” controlled polymerization in a one-pot, two-step process as reported in Figure 16A. These copolymers were based on a graft structure, which typically leads to superior physical and mechanical properties compared to linear block copolymers. As shown in Figure 16B, the oxygen barrier properties of the PLA-based graft copolymer films were evaluated and compared to those of commercial PLA and poly(styrene)−b−poly(isoprene)−b−poly(styrene) (SIS), which is one of the most widely used TPEs in food packaging applications [274]. Depending on compositions, the resulting superelastomers showed high-performance gas barrier characteristics. The oxygen permeability performance of copolymers with the highest PLA content (45 wt%) was 60-fold and 5-fold better than that of SIS and neat PLA, respectively. This was attributed to the presence of the largest fraction of homogeneously distributed semicrystalline PLA domains, which tied up the rubbery isobutylene segments, thus decreasing the channel for gas permeability through the copolymer matrix.

## 7. Molecular Dynamics Simulations for Permeability Investigation of PLA-Based Materials

### 7.1. Most Important Tools to Study PLA-Based Plastic

In line with the growing computing power and with an ever increased number of precise and rigorous programs available to the scientific community, there have been several proposed theoretical models for the prediction of polymer permeability that are discussed in this section. Among the available approaches, molecular dynamics (MD) simulations have been successfully applied in different research areas [279] including mass transport property investigations to elucidate the sorption and diffusion mechanisms of small gas molecules in varieties of potentially useful polymers such as PLA [280]. Applying these methods, many results, difficult or impossible to detect from conventional experiments, can be obtained.

Different MD approaches can be used to investigate the permeability of PLA-based materials. Sun and Zhou [281] performed full-atomistic simulations of oxygen sorption and diffusion in amorphous PLA. The oxygen solubility coefficient (*S*) was calculated by fitting the dual-mode sorption model to the simulation data. The simulated *S* value was much higher than the experimental data obtained from the time-lag method, but slightly lower than the measurement from quartz crystal microbalance. This discrepancy was probably due to the predominant Langmuir type sorption mechanism, which holds for oxygen sorption in PLA. The time-lag method only considers oxygen molecules that are involved in the diffusion process. The allowed rotation states of successive bonds between adjacent atoms are determined from probability functions by energy consideration using the standard Monte Carlo method [282]. The solubility coefficient of gas in a glassy polymer is defined as the ratio of gas concentration to gas pressure at equilibrium, following Equation (4):(4)$$S=\frac{C}{p}=k_{D}+{C_{H}}^{\prime }\;\frac{b}{1+bp}$$
where *k*_D_, C_H_′, and *b* can be determined by nonlinear regression fit of the oxygen sorption data that are obtained by GCMC simulations.

Information about the structural features of different PLA models at the atomic level can be provided by the radial distribution function (RDF). The RDF indicates the probability density of finding *A* and *B* atoms at a distance of *r* following Equation (5):(5)$$g_{A-B}(r)=\left(\frac{n_{B}}{4\pi r^{2}dr}\right)/\left(\frac{N_{B}}{v}\right)$$
where *n_B_* is the number of *B* particles located at the distance *r* in a shell of thickness *dr* from particle *A*; *N_B_* is the number of *B* particles in the system; and *v* is the total volume of the system. The free volume inside the polymer matrix can be obtained by rolling a spherical probe over the Connolly surface of polymer atoms. It should be noticed that the available volume for the probe to pass through is dependent on the radius of the probe [78]. Regardless of the specific computational type to use, the diffusivity of a gas in an organic solvent, polymer, or zeolite can be calculated by running a MD simulation and determining the mean square displacement (MSD) of the gas in the material. MSD is a measure of the deviation of the position of a particle with respect to a reference position over time. It is the most common measure of the spatial extent of random motion, and for this reason, it is the most useful tool to detect the small molecule diffusion from a starting state [283]. The motion pattern of penetrant gases in the host polymer can be qualitatively studied by monitoring the penetrant’s displacement |*r*(*t*) − *r*(0)| from its initial position. The diffusion coefficient D is then obtained from the slope of a plot of the MSD against time t.

H. Ebadi-Dehaghani et al. used a MD simulation to investigate and predict the gas permeability through polymer blends and nanocomposites [284]. They demonstrated that the oxygen permeation was highly dependent on blend composition, clay loading, and state of clay dispersion governed by compatibilization in the PP/PLA/clay nanocomposite film. Compared to the PLA-rich system, they found that the PP-rich films showed a greater barrier to oxygen. The lower permeability of PP-rich films was mainly due to the reduced size of dispersed domains of the oxygen barrier component (i.e., PLA in the PP matrix), leading to a more tortuous permeant path, while the higher degree of crystallinity observed in PP-rich films compared to the PLA-rich system was found to also be responsible for the higher barrier properties of the PP-rich films.

### 7.2. Different MD Approaches to Use for PLA Systems

The use of explicit full-atom (FA) simulation model (Figure 17A) for PLA polymers has been found to be suitable for approximately reproducing different important physical properties of amorphous PLA solids. In particular, Xiang and Anderson adopted this kind of approach to calculate material density, water sorption isotherm, and diffusion coefficient of PLA systems, thus verifying potential utilities in designing PLA based drug delivery systems, particularly for predicting drug–PLA miscibility. They combined MD simulations, the particle insertion method of Widom [285], and a theoretical sorption relation to calculate the water sorption isotherm in PLA. They found that at 0.6 (wt%) of H_2_O, water molecules localized next to polar ester groups in PLA because of hydrogen bonding. Local mobility in PLA as characterized by the atomic fluctuation was sharply reduced near the Tg, decreasing further with aging at 298 K [286]. The non-Einsteinian diffusion of water was found to be correlated with the rotational β-relaxation of PLA carbonyl groups at 298 K. A relaxation–diffusion coupling model proposed by the authors provided a diffusion coefficient of 1.3 × 10^−8^ cm^2^/s at 298 K, which is comparable to the reported experimental values [287]. In other studies, MD simulations have been performed to estimate the diffusivity coefficients of the gases CO_2_, O_2_, and N_2_ from polypropylene (PP)/poly(lactic acid) (PLA)/clay nanocomposite films with various compositions (PP-rich and PLA-rich). Diffusion temperature dependency of the PP-rich sample for O_2_ gas has been also investigated. The MSD of the gases has been calculated via MD simulation according to the Charati and Stern method [288], which consists of the generation of the initial microstructure of a polymer containing penetrant gas molecules as an amorphous cell module, and the use of a full atom approach. The diffusion coefficients of gases in PLA can also be controlled by the amount of free volume, the free-volume distribution, and the dynamics of the free volume of the polymers. Penetrant molecules reside most of the time in microcavities inside the PLA matrix, and the microcavities are elements of “free volume” (or “empty” volume) between the surrounding polymer chains. This reliable computational method showed that solubility increased with increasing temperature, which was in accordance with the experiments.

Another explicit FA computational model for PLA was developed by Xiang and Anderson (2014). MD simulations of PLA glasses were carried out to explore various molecular interactions and predict certain physical properties such as material density, water sorption isotherm, and diffusion coefficient. The combined use of MD simulations, the particle insertion method of Widom, which is a statistical thermodynamic approach to the calculation of material and mixture properties [285], and a theoretical sorption relation allows us to efficiently calculate the water sorption isotherm in PLA. Weak sorption of water in amorphous PLA solids can be predicted, with results that can be similar to the experimental ones. Inspection of molecular structures of the simulated PLA glasses provided further understanding of the distribution of water in PLA polymers, which has been difficult to obtain experimentally.

The FA MD studies of fundamental properties such as water solubility, diffusivity, and distribution in PLA polymers are only a few, due to the enormous computational resources required to conduct atomistic simulations with explicit solvent models. United atom (UA) MD has also been proposed to obtain cheaper calculations than those obtained for all atom MD, retaining a high accuracy degree. This method subsumes nonpolar hydrogen atoms into their adjacent carbon atom (Figure 17B), decreasing the computational costs. While the accuracy of the united atom MD simulation has been found to be reliable for protein modeling [289], this approach was found to be unreliable in estimating the diffusion coefficients of small penetrant molecules through polymers.

Coarse-grained (CG) modeling represents a valid way to overcome the huge requirements for computing resources while maintaining high calculation efficiency. It consists of a less sensitive method than the FA and UA approaches (Figure 17C), but its accuracy degree remains high enough to describe short- and long-range phenomena at various granularity levels [290,291]. Two CG computational methods can be considered for the investigation of PLA-based material permeability: a mesoscale approach—dissipative particle dynamics (DPD) [292]—and MD simulations using the MARTINI force field in conjunction with the GROMACS package [293]. DPD simulations group atoms and molecules into fluid beads and use bead level interactions to describe the evolution of a system. For any two beads *i* and *j*, the pair wise interaction force (F^DP^*_ij_*) is the sum of the conservative force (F^C^*_ij_*), dissipative force (F^D^*_ij_*), and random force (F^R^*_ij_*), as shown in Equation (6):F^DP^*_ij_* = F^C^*_ij_* + F^D^*_ij_* + F^R^*_ij_*(6)
where F^C^*_ij_* is a soft repulsive force, while F^D^*_ij_* is a drag force or frictional force and F^R^*_ij_* is a random force. Of these three, the conservative force (F^C^*_ij_*) best describes the energy of the system. Groot and Warren (1997) established a connection between DPD beads and a real fluid by defining a relationship between the maximum repulsion between particles (a*_ij_*), which is a function of the conservative force (F^C^*_ij_*), and the Flory–Huggins interaction parameter (*x*). The MARTINI force field uses, on average, a four to one mapping of non-hydrogen atoms to interaction centers (although sometimes fewer or more than four atoms are mapped on to an interaction site) and defines the interaction sites into four main types: polar (neutral water soluble atoms), non-polar (mixed groups of polar and apolar atoms), apolar (hydrophobic groups), and charged (groups bearing an ionic charge) [294].

FA and CG MD simulations represent different ways to carry out MD simulations for the permeability investigation of PLA-based materials. There is no specific method to conduct this as the choice to use one protocol with respect to the other depends on the size of the conditions of the system (PLA size, presence of solvent, number of gas molecules, and simulation time). Recent studies [295] have shown that it is possible to use both methods. This combined approach is based on starting from an all-atom MD simulation to obtain input parameters such as angles, bonds, and dihedrals of PLA chains, taking into account the crystalline and amorphous phases. After that, the MARTINI force field could be used to map PLA with various polymer segment lengths against the presence or absence of other molecules in the system.

## 8. Concluding Remarks and Future Outlook

With the aim of minimizing the environmental impact, several PLA-based biodegradable plastic packaging materials with improved barrier properties have been developed in the last decades. In this work, all recent strategies to improve the barrier properties of PLA were reviewed including crystallization, orientation, stereocomplexation, blending, incorporation of nanoparticles, and copolymerization. Considering the enormous number of available approaches and the wide range of processing conditions applied, it is often difficult to establish clear relationships between the structure and barrier properties. However, the current review of the literature has identified some strategies that appear to have a greater impact on the gas permeability of PLA than others.

Starting with crystallinity content, although it has been generally demonstrated that barrier properties depend on the degree of crystallinity for many semi-crystalline polymers, increasing the crystallinity content of PLA from 0 to 40% does not always lead to a decrease in gas permeability, as expected. For PLA materials with a comparable degree of crystallinity, several studies have shown that gas permeability strongly depends on chain orientation, amorphous phase morphology, and crystalline forms (i.e., ordered α-form crystals and less ordered α′-form crystals in PLA). More importantly, further studies on the wide variety of crystalline and amorphous phase organization of PLA have revealed the importance of the crystalline superstructure and lamellar arrangement in improving gas barrier properties. In other words, the crystalline phase acts as an impermeable barrier, but its supermolecular microstructure must be properly tailored. Numerous data show that gas barrier properties can be reduced by up to two orders of magnitude if the arrangement of PLA lamellae is tuned along a perpendicular direction of the gas diffusion path. To achieve this, specific crystallization protocols must be applied to create regularly spaced lamellae in PLA packaging materials. The most successful approaches involve either molecular orientation, crystallization under nanoconfinement, addition of vertically aligned nanoplatelets, and/or use of specific fibrillar nucleating agents as templates to construct parallel-aligned shish-kebab-like crystals with well-interlocked boundaries. Consistent with these promising findings, most recent studies have focused on the development of nanocomposite structures by incorporating small amounts of nanoparticles (i.e., typically clays with at least one dimension less than 100 nm) into polymeric matrices. By acting as physical barriers to the diffusion and permeation of small molecules, such nanostructures have been shown to significantly improve the barrier properties of PLA. This phenomenon is attributed to the well-known tortuosity effect. In this context, theoretical models and empirical studies have suggested that, in addition to the aforementioned nanoparticle orientation, other key factors in achieving high-barrier materials are maximizing the aspect ratio of the nanoparticles and improving the interfacial adhesion between the nanofillers and the PLA matrix. However, despite extensive research efforts, none of these materials, with the level of gas permeability required by the end-users and at reasonable costs, have reached the market yet. As this film will be in contact with food, potential risks to human health are questioned as nanomaterials may migrate from the packages to foodstuff. In addition, the effect of nanoparticle incorporation on the biodegradability/compostability of the polymer must be thoroughly investigated. In the absence of these detailed studies, nanocomposites cannot be approved for legal utilization in industry.

Meanwhile, recent progress in polymerization methods such as controlled radical polymerization (CRP) has enabled the synthesis of interesting copolymerization products for a wide range of applications, while fitting with some of the principles of green chemistry such as the use of bio-based monomers, solvent-less protocols, and mild conditions (low temperature). Using these methods, PLA barrier properties can be enhanced by copolymerization with appropriate monomers to widen its applicability in the food packaging field. Despite the impressive developments, there are still challenges that need to be addressed. For instance, there is a huge need to discover new selective catalysts in light of expanding the range of monomers that can be used. In addition, current polymerization protocols are difficult to scale-up due to the lack of automatic systems. However, further developments are expected in the near future as growing research interest is focusing on strategies to simplify the current polymerization protocols and allow for more scalable methodologies to be implemented.

## Figures and Tables

**Figure 1 polymers-14-01626-f001:**
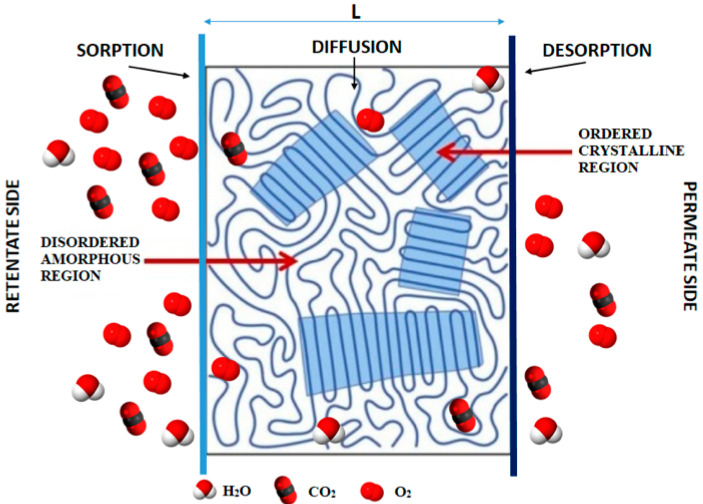
Schematic representation of the general mechanism of the permeation of small molecules through semicrystalline polymers.

**Figure 2 polymers-14-01626-f002:**
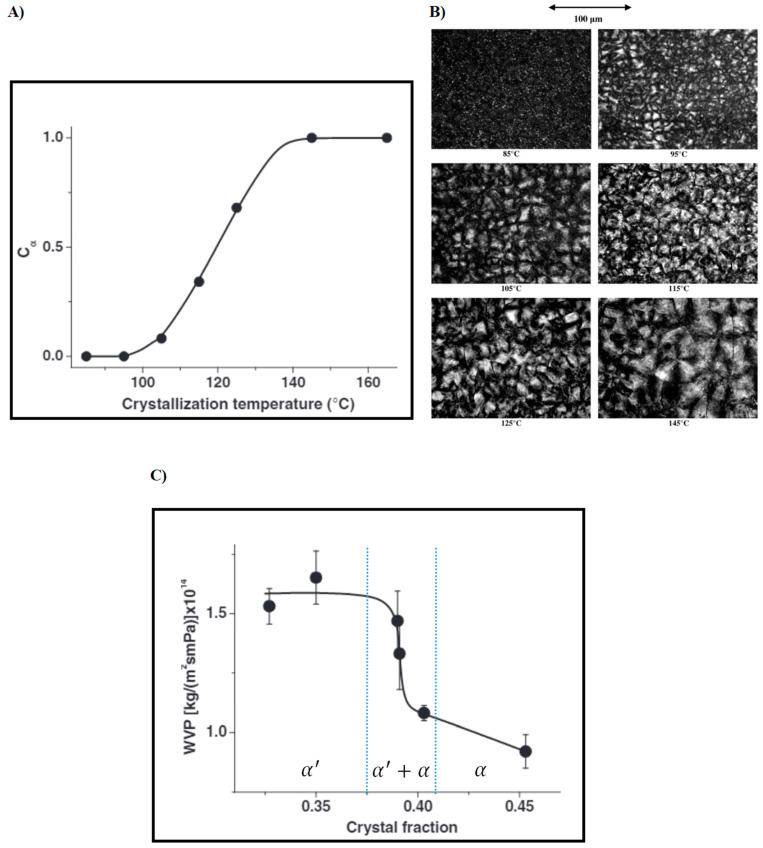
(**A**) Water vapor permeation estimated fraction of the α form in PLLA films; (**B**) optical micrographs of compression molded PLLA films after cold crystallization; (**C**) water vapor permeability of PLLA films crystallized as a function of degree of crystallinity. Adapted from [83] with permission from Elsevier. Copyright © 2011.

**Figure 3 polymers-14-01626-f003:**
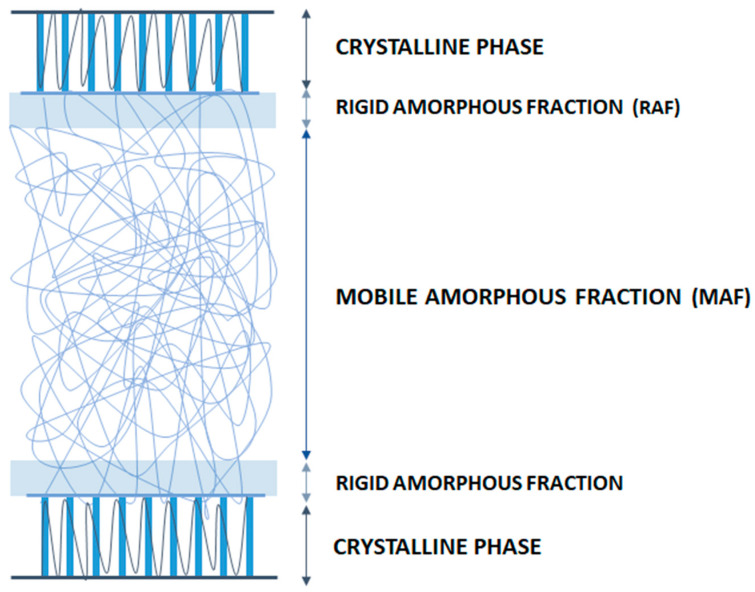
Schematic representation of the arrangement of crystalline, rigid amorphous and mobile amorphous fractions.

**Figure 4 polymers-14-01626-f004:**
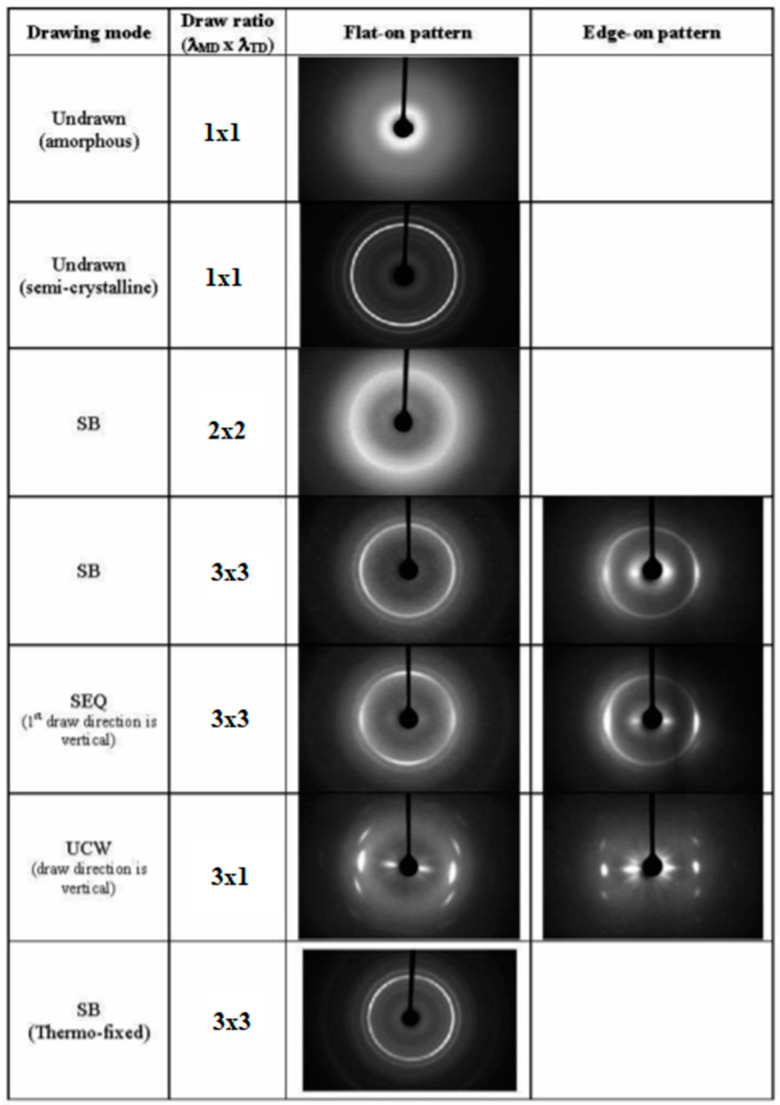
WAXD patterns of drawn and thermally crystallized PLA films. Adapted from [62] with permission from American Chemical Society. Copyright © 2012.

**Figure 5 polymers-14-01626-f005:**
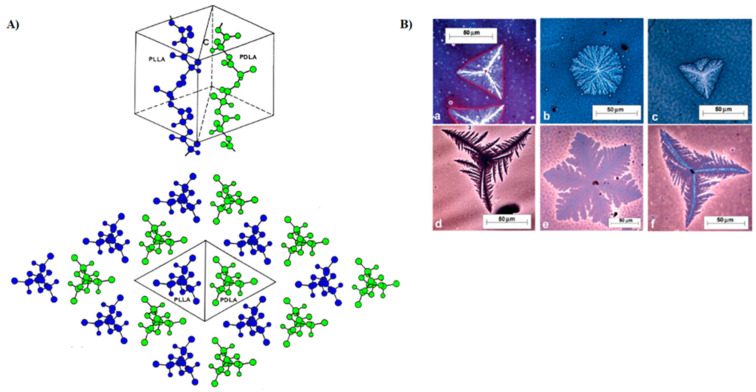
(**A**) SC crystalline lattice; adapted from [115] with permission from Elsevier. Copyright© 2016. (**B**) Optical microscopy micrographs of different PLLA/PDLA films with PLLA content of 75% (**a**,**d**), 50% (**b**,**e**), and 25% (**c**,**f**), crystallized at 200 °C; reproduced from [116] with permission from American Chemical Society. Copyright © 2010.

**Figure 6 polymers-14-01626-f006:**
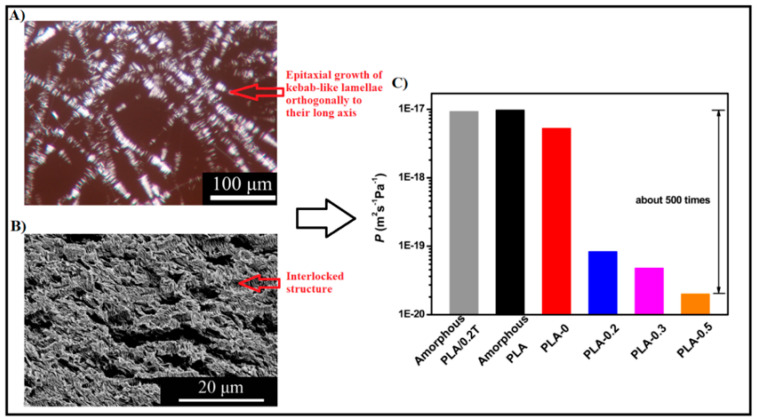
(**A**,**B**) POM and cross-sectional micrograph of crystal morphology for PLA with a nucleating agent showing an epitaxial growth lamellae and interlocked structure, respectively; (**C**) Oxygen permeability coefficient (*p*) values for all PLA samples as a function of nucleating agent content. Adapted from [125] with permission from American Chemical Society. Copyright © 2014.

**Figure 7 polymers-14-01626-f007:**
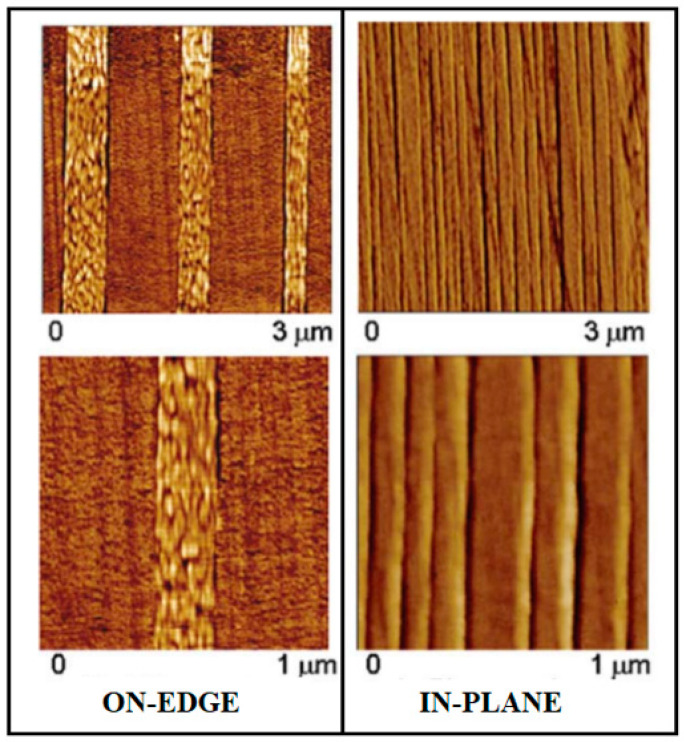
Example of typical AFM phase images of multilayered film cross-section with on-edge and in-plane orientations. Adapted from [133] with permission from John Wiley and Sons. Copyright © 2011.

**Figure 8 polymers-14-01626-f008:**
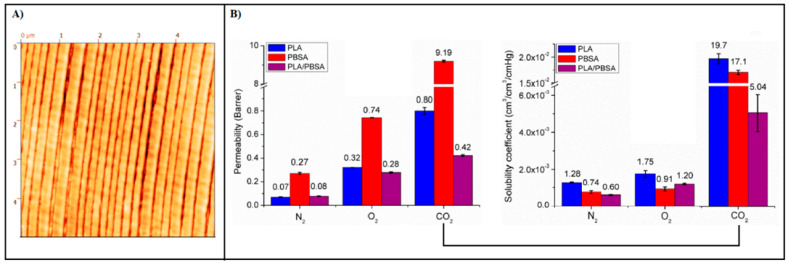
(**A**) AFM images of the multilayer film with PLA in light and PBSA in dark; (**B**) gas permeability and solubility coefficients for the monolayer films of PLA and PBSA and the multilayer film of PLA/PBSA. Adapted from [137] with permission from American Chemical Society. Copyright © 2017.

**Figure 9 polymers-14-01626-f009:**
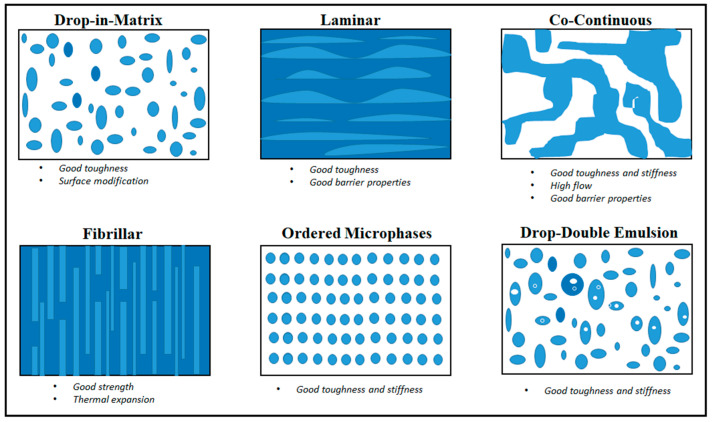
Types of morphology in immiscible polymer blends.

**Figure 10 polymers-14-01626-f010:**
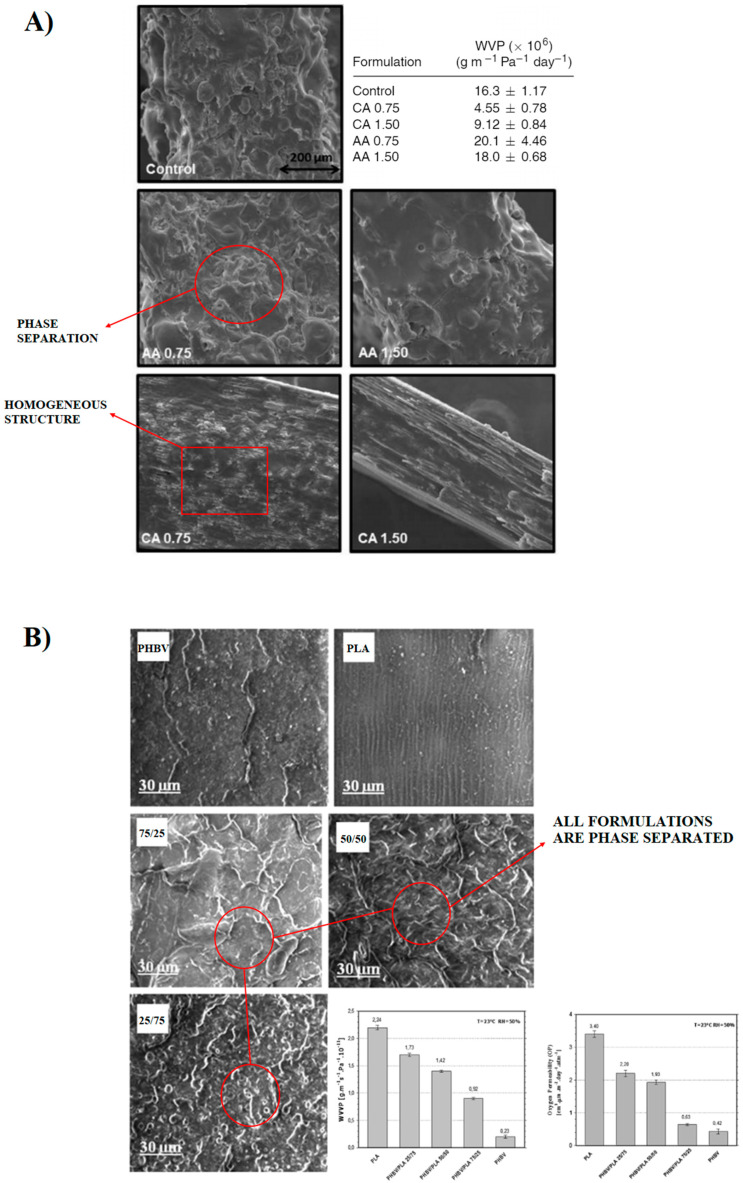
SEM images of (**A**) PLA/TPS sheets with CA (including WVP values), adapted from [191] and (**B**) fracture surface of neat PHBV, neat PLA, and their blends (including WVP and oxygen permeability values), adapted from [193] with permission from Elsevier. Copyright © 2013.

**Figure 11 polymers-14-01626-f011:**
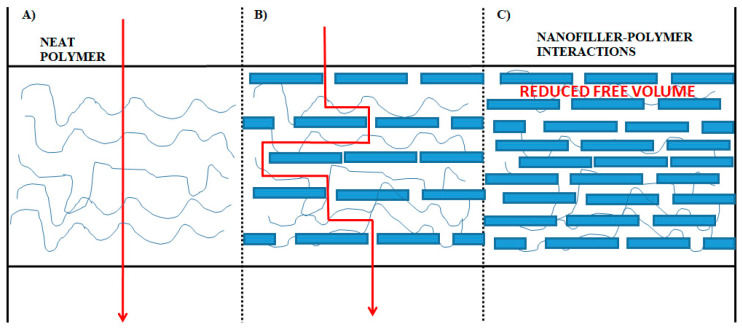
Simplified drawing of the “tortuous path” produced by the incorporation of exfoliated clay nanoparticles into a polymer matrix. (**A**) Neat polymer (diffusing gas molecules follow a pathway perpendicular to the film orientation); (**B**) non-interacting nanocomposite (impermeable platelets hinders direct diffusion); (**C**) interacting nanocomposite (the polymer strands are “immobilized” at the polymer–nanofiller interface and the overall free volume available is reduced.

**Figure 12 polymers-14-01626-f012:**
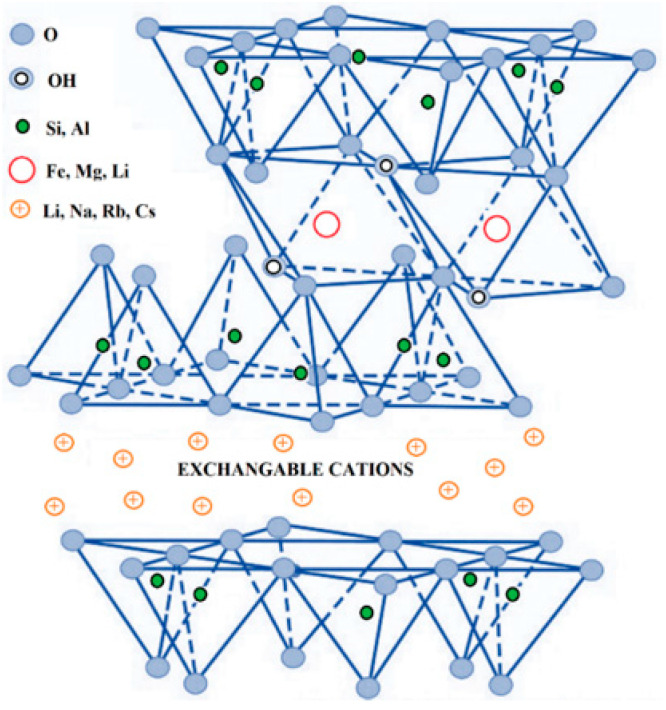
Example of the typical structure of the 2:1 layered silicates. Redrawn from [212] with permission of Elsevier. Copyright © 2015.

**Figure 13 polymers-14-01626-f013:**
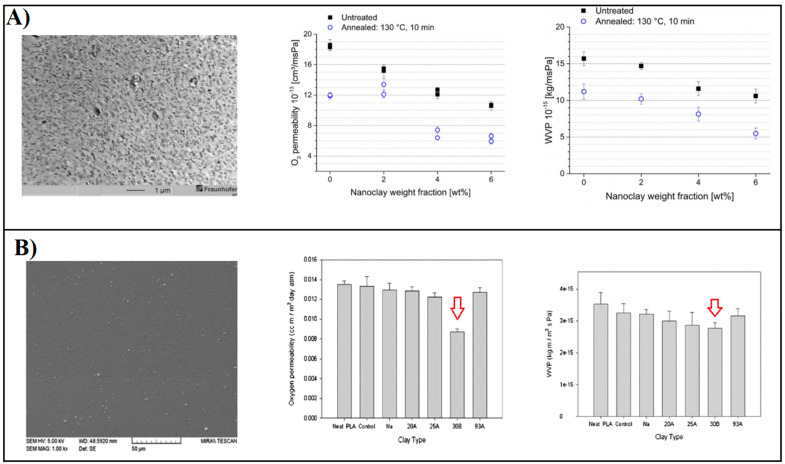
From left to right: (**A**) TEM image of PLA based nanocomposite films containing 6 wt% CH 30B along with oxygen and WVP values of untreated nanocomposites and annealed at 130 °C for 10 min as a function of nanoclay content. Adapted from [231] with permission of Wiley Periodicals. Copyright © 2016. (**B**) TEM and SEM images of PLA-CH 30B nanocomposites along with the relative oxygen permeability of nanocomposites with different volume fraction of clays (CNa:CH Na^+^, CRDP:Fyrolflex, and C30B:CH 30B). Adapted from [234] with permission from Elsevier. Copyright © 2016.

**Figure 14 polymers-14-01626-f014:**
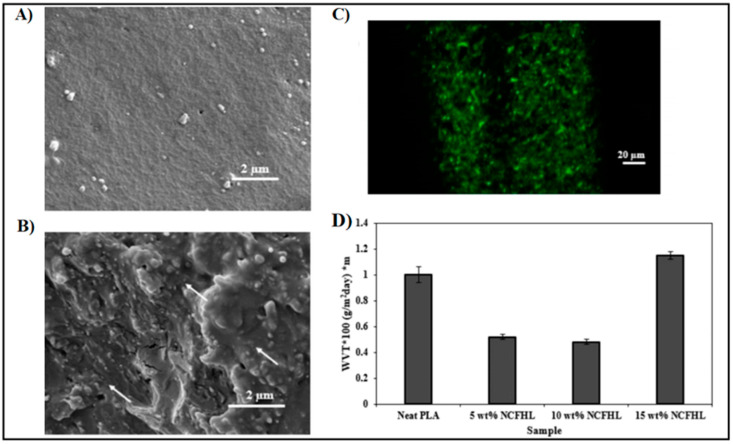
TEM image of (**A**) neat PLA and (**B**) the PLA–NCFHL biocomposite (with lignin) in which fibrils were well embedded within the PLA matrix (indicated by arrows); (**C**) confocal laser microscope image showing well dispersed lignin fibrils on the PLA surface; (**D**) Water vapor transmission rate of the neat PLA and PLA/lignin biocomposite. Adapted from [248] with permission from American Chemical Society. Copyright © 2018.

**Figure 15 polymers-14-01626-f015:**
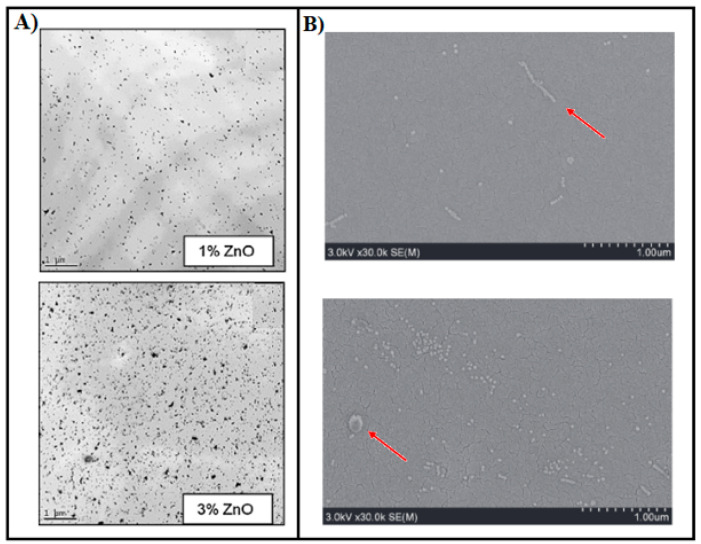
(**A**) TEM images of surface coated PLA nanocomposites filled with 1 and 3 wt% of ZnO nanoparticles (even dispersion), adapted from [262] with permission from Elsevier. Copyright© 2013. (**B**) SEM images of PLA nanocomposites containing 1–1.5 wt% of ZnO without any surface treatment (particle agglomeration indicated by red arrows), adapted from [158] with permission of Elsevier. Copyright © 2018.

**Figure 16 polymers-14-01626-f016:**
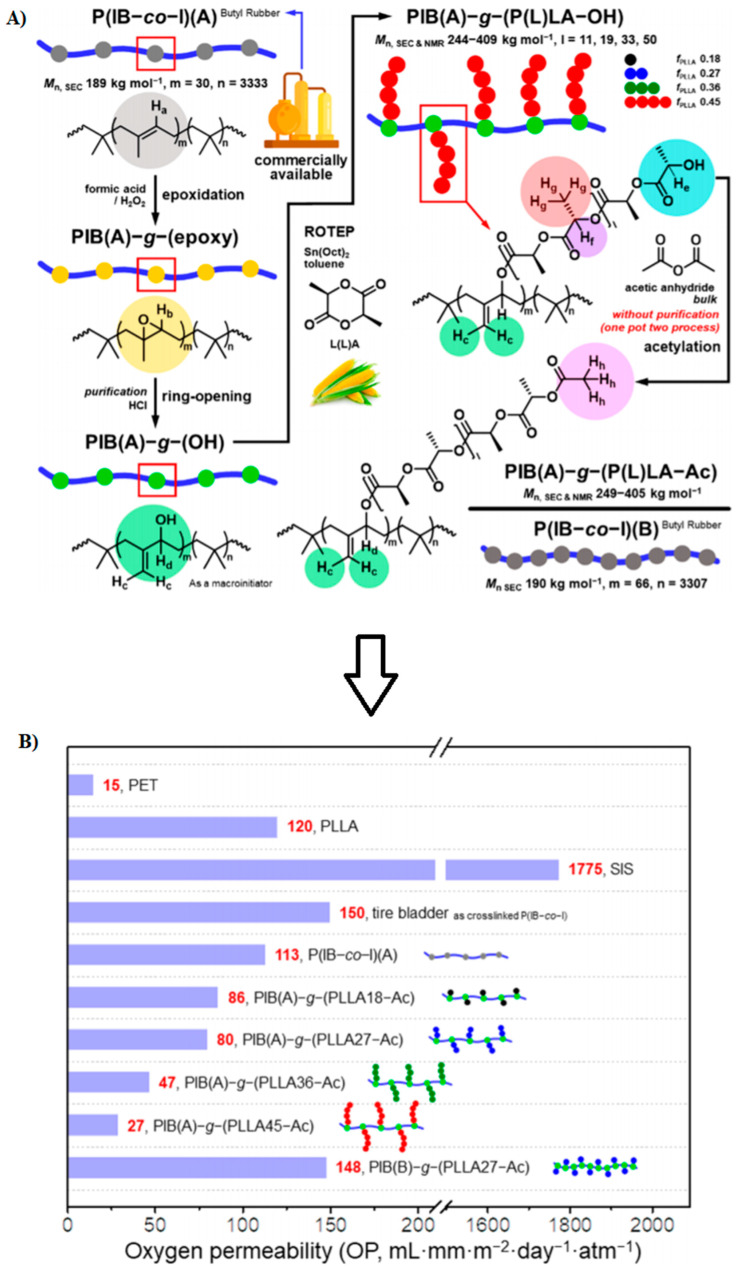
(**A**) Schematic synthesis of semicrystalline or amorphous poly(isobutylene)-graft-acetylated poly(lactide) (PIB-g-(P(L)LA−Ac)). (**B**) Comparison of oxygen permeability values between the newly synthesized polymers and common ones. Adapted from [274] with permission from American Chemical Society. Copyright © 2020.

**Figure 17 polymers-14-01626-f017:**
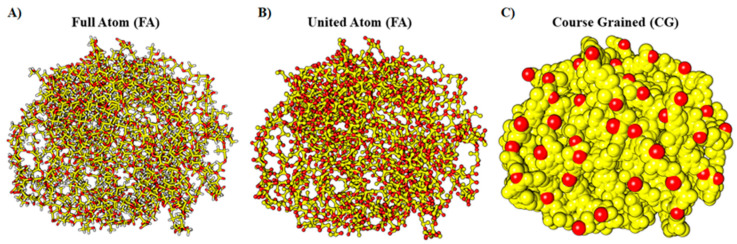
Representations of PLA nanoparticles using (**A**) full atoms (FA), (**B**) united atoms (UA), and (**C**) course grained (CG) methods.

**Table 2 polymers-14-01626-t002:** Water vapor permeation coefficient (P_water_) of PLA films measured at 25 °C as a function of degree of crystallinity ($X_{c}$) and crystallization time. Data taken from [78].

$X_{L}{\;}^{a}\;(\%)$	Crystallization Time ^b^ (min)	$X_{C}{\;}^{c}\;(\%)$	P × 10^14^ (Kg m/m^2^/s/Pa)
72.2	0	0	1.90
50	0	0	1.95
99.7	0	0.6	2.08
99.4	0	0.7	2.18
99.4	0	1	1.91
99.4	5	5.7	1.90
99.4	7.5	19.1	1.14
99.4	10	32	0.99
99.4	12.5	34.9	1.04

^a^ L-lactyl unit content $X_{L}$ of P(LLA-DLA). ^b^ Crystallization time equal to 0 means that the specimens were melt-quenched. ^c^ Crystallinity content obtained from the DSC first run.

**Table 3 polymers-14-01626-t003:** Values related to samples’ crystallinity degree ($X_{c}$), mobile amorphous phase degree ($X_{am}$), rigid amorphous fraction degree ($X_{ar}$) as well as permeability coefficient P. Data taken from [62].

PLA Sample	$X_{C}{\;}^{a}\;(\%)$	$X_{am}{\;}^{b}\;(\%)$	$X_{ar}{\;}^{c}\;(\%)$	P_water_ ^d^
Amorphous film	0	100	0	2.15
Thermally crystallized film	31	41	28	2.04
UCW drawn (3 × 1)	28	66	6	2.04
SEQ drawn (3 × 3)	27	70	3	1.97
SB drawn (2 × 2)	13	86	1	1.76
SB drawn (3 × 3)	25	66	9	1.63
SB drawn (4 × 4)	31	62	7	1.63
SB drawn (3 × 3) thermos-fixed	31	59	10	1.70

^a^ Crystallinity degree. ^b^ Mobile amorphous phase degree. ^c^ Rigid amorphous fraction degree. ^d^ Water permeability coefficient (10^−12^ mol m^−1^ s^−1^ Pa^−1^).

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
