# Peer review of "Tailoring the Barrier Properties of PLA: A State-of-the-Art Review for Food Packaging Applications"

_polymers, 2022, doi:10.3390/polym14081626_

Round 1

Reviewer 1 Report

Some concepts should be synthesized in a more concise way – e.g. lines 827 -939: Layered nanofillers; lines 1014-1084: Isodimensional nanoparticles

Acronyms should be defined the first time when they are introduced –

Some examples (not a complete list):

Line 72: PET, PVOH

Table 1 all names of the polymers should be introduced in addition to their acronyms

Line 512: POM

Line 636: PCL, PBS, PHAs

Line 671: SEM

In addition, acronyms should be carefully checked to be consistently used in the same format throughout the whole review.

Typos/errors should be fixed:

Some examples (not a complete list)

Line 157: “he”

Line 637:”nanowishers”

Line 973:” compatibilitazion”

Line 1009:” likely due strong”

Lines 1014, 1016:” Isodimentional”

Lines 1040-1041:” The different barrier behaviors observed among samples was linked to…”

------------------------

All Figures and Tables summarizing literature data should include citation as the text in the main body of the manuscript

Figure 9. caption should be placed below the figure  as in other figures in the text.

Lines 697 “water permeability was reduced” and 700 “water permeability coefficients were found to increase,” do not seem to be well connected with line 698 “In line with these results”

Lines 198-200: Crystallinity of EVOH and PVDC should be defined as they are described as “highly crystalline materials;” since crystallinity of polymers is provided in Table 1, it would be good to provide this information  for EVOH and PVDC for comparative purposes.

Lines 224-226: This statement does not seem to refer to data presented in Table 2

Lines 742 – 754: Effect if cinnamaldehyde on WVP and OP is not discussed but only its effect on mechanical properties (“in the absence of compatibilizer” decrease in WVP and OP; what were the results in the presence of compatibilizer?)

Lines 778-780:”—water and oxygen barrier properties-----reaching a reduction of about 75% and 81.5% respectively” At what PHBV/PLA ratio were these reduction rates achieved?

Line 793: Are these phenomena characteristic strictly for PLLA?

Lines 976-977 :” Since lignin contains large amounts of nonpolar hydrocarbon chains” this statement is not quite correct.

References should be checked: 

For example, some references are not of the same format - e.g. #45 and #48, while #59, #90, #93 are not complete

Author Response

Reviewer 1

Thank you very much for all your valuable comments. Each of your insights have served to strengthen our manuscript.

  1. Some concepts should be synthesized in a more concise way – e.g. lines 827 -939: Layered nanofillers; lines 1014-1084: Isodimensional nanoparticles

The related parts have been modified accordingly.

  1. Acronyms should be defined the first time when they are introduced –

Some examples (not a complete list):

Line 72: PET, PVOH

Table 1 all names of the polymers should be introduced in addition to their acronyms

Line 512: POM  

Line 636: PCL, PBS, PHAs

Line 671: SEM

In addition, acronyms should be carefully checked to be consistently used in the same format throughout the whole review.

Acronyms have been checked and corrected throughout the whole paper.

  1. Typos/errors should be fixed:

Some examples (not a complete list)

Line 157: “he”

Line 637:”nanowishers”

Line 973:” compatibilitazion”

Line 1009:” likely due strong”

Lines 1014, 1016:” Isodimentional”

Lines 1040-1041:” The different barrier behaviors observed among samples was linked to…”

All typos/errors have been checked carefully again and corrected.

  1. All Figures and Tables summarizing literature data should include citation as the text in the main body of the manuscript

This has been carefully checked and corrected.

  1. Figure 9. caption should be placed below the figure  as in other figures in the text.

Adjusted.

  1. Lines 697 “water permeability was reduced” and 700 “water permeability coefficients were found to increase,” do not seem to be well connected with line 698 “In line with these results”.

Corrected.

  1. Lines 198-200: Crystallinity of EVOH and PVDC should be defined as they are described as “highly crystalline materials;” since crystallinity of polymers is provided in Table 1, it would be good to provide this information  for EVOH and PVDC for comparative purposes.

This information has been added as suggested.

  1. Lines 224-226: This statement does not seem to refer to data presented in Table 2

Corrected.

  1. Lines 742 – 754: Effect if cinnamaldehyde on WVP and OP is not discussed but only its effect on mechanical properties (“in the absence of compatibilizer” decrease in WVP and OP; what were the results in the presence of compatibilizer?)

The missing information has been added.

  1. Lines 778-780:”—water and oxygen barrier properties-----reaching a reduction of about 75% and 81.5% respectively” At what PHBV/PLA ratio were these reduction rates achieved?

The missing information has been added.

  1. Line 793: Are these phenomena characteristic strictly for PLLA?

In the packaging field, only PLA grades with L-lactide contents of about 95.75 and 98.6 weight% (PLLA) are useful thanks to their better mechanical/optical properties and slower degradation rate than poly(d,l-lactic acid) (PDLLA) due to crystalline regions. Therefore, the majority of works in this field has been conducted on PLLA.

  1. Lines 976-977 :” Since lignin contains large amounts of nonpolar hydrocarbon chains” this statement is not quite correct.

The sentence has been made clearer.

  1. References should be checked: 

For example, some references are not of the same format - e.g. #45 and #48, while #59, #90, #93 are not complete

The references have been carefully checked and their format corrected.

Reviewer 2 Report

This Review is exceptional. I found it to be very well structured, informative, and should be accepted for publication. One addition that would be helpful is the data for PHB and PHV to Table 1. These are discussed in the blending section and so having the pure polymer properties in Table 1 would be helpful.

Author Response

Reviewer 2

This Review is exceptional. I found it to be very well structured, informative, and should be accepted for publication. One addition that would be helpful is the data for PHB and PHV to Table 1. These are discussed in the blending section and so having the pure polymer properties in Table 1 would be helpful.

Thank you very much for your valuable comments and feedback regarding our review paper. We added the data on PHBV in Table 1 as suggested.
